# GENOME: GENERATIVE NEURO-SYMBOLIC VISUAL REASONING BY GROWING AND REUSING MODULES

**Zhenfang Chen** [*]
MIT-IBM Watson AI Lab

**Rui Sun**[*]
Columbia University

**Wenjun Liu**[*]
Tsinghua University

**Yining Hong**
University of California, Los Angeles

**Chuang Gan**
MIT-IBM Watson AI Lab and UMass Amherst

## ABSTRACT

Recent works have shown that Large Language Models (LLMs) could empower traditional neuro-symbolic models via programming capabilities to translate language into module descriptions, thus achieving strong visual reasoning results while maintaining the model's transparency and efficiency. However, these models usually exhaustively generate the entire code snippet given each new instance of a task, which is extremely ineffective. On the contrary, human beings gradually acquire knowledge that can be reused and grow into more profound skills for fast generalization to new tasks since we are an infant. Inspired by this, we propose generative neuro-symbolic visual reasoning by growing and reusing modules. Specifically, our model consists of three unique stages, module initialization, module generation, and module execution. First, given a vision-language task, we adopt LLMs to examine whether we could reuse and grow over established modules to handle this new task. If not, we initialize a new module needed by the task and specify the inputs and outputs of this new module. After that, the new module is created by querying LLMs to generate corresponding code snippets that match the requirements. In order to get a better sense of the new module's ability, we treat few-shot training examples as test cases to see if our new module could pass these cases. If yes, the new module is added to the module library for future reuse. Finally, we evaluate the performance of our model on the testing set by executing the parsed programs with the newly made visual modules to get the results. We find the proposed model possesses several advantages. First, it performs competitively on standard tasks like visual question answering and referring expression comprehension; Second, the modules learned from one task can be seamlessly transferred to new tasks; Last but not least, it is able to adapt to new visual reasoning tasks by observing a few training examples and reusing modules[1].

## 1 INTRODUCTION

Neuro-symbolic visual reasoning models (Andreas et al., 2016b; Mao et al., 2019b) refer to the algorithm family that combines deep neural networks (lec, 1998; Hochreiter & Schmidhuber, 1997) for learning correlations among the training data and symbolic methods (Yi et al., 2018; Andreas et al., 2016a) to perform explicit and transparent multi-step reasoning. In contrast to pure neural network-based models (Hudson & Manning, 2018; Li et al., 2023), neuro-symbolic approaches achieve strong performance in visual reasoning tasks, simultaneously offering superior model transparency and data efficiency.

Nevertheless, such models suffer from several inherent limitations. Firstly, their language parsers (Yi et al., 2018; Andreas et al., 2016b), employed for the conversion from natural language into symbolic programs, typically demand extensive domain-specific language-program pairs to train on, and struggle to generalize effectively to unconstrained natural language instructions. Additionally, these models necessitate a custom design for every module, rendering the process labor-intensive and lacking scalability.

---

[*]indicates equal contributions
[1]Project page: https://vis-www.cs.umass.edu/genome

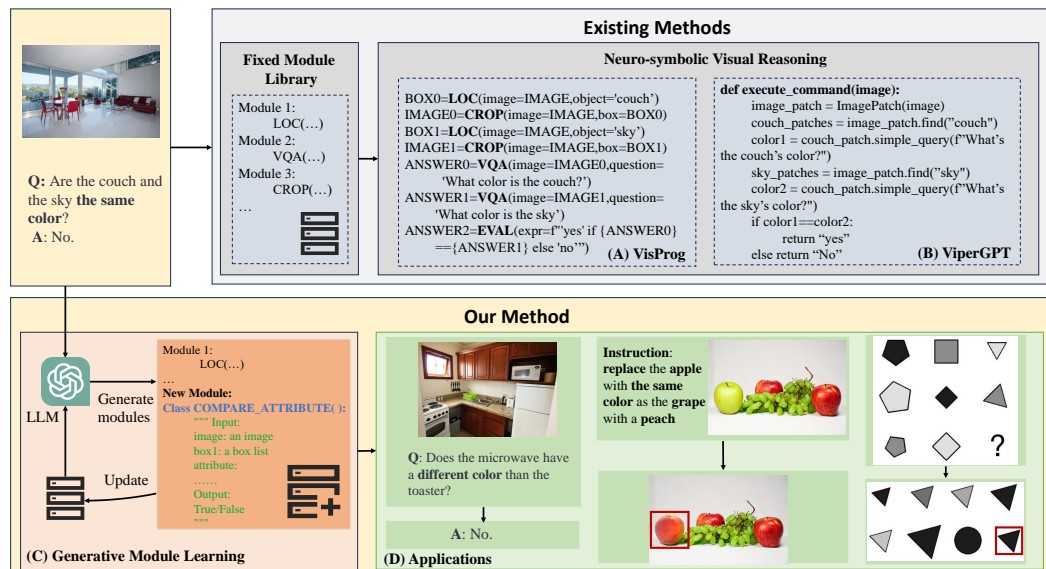

Figure 1: The motivation of the GENOME. Compared with VisProg and ViperGPT which exhaustively generate a code snippet for each input case, our GENOME is able to generate new modules and reuse old modules to handle the query. The module generated by GENOME can be used to handle other instances of the task for better performance. Second, the generated module can be transferred to different tasks like image editing. Finally, it can learn to handle new tasks like Raven (Burke, 1985; Zhang et al., 2019a) by learning modules from only a few training samples. The edited region and the correct answer for the Raven task are labeled with red boxes for better visualization.

Recent advancements in large language models (LLMs) (Brown et al., 2020; Ouyang et al., 2022) have ushered in a new era with its remarkable performances across various applications, including chatbots (Shuster et al., 2022), virtual assistants (Dong et al., 2023), and programming assistants (Chen et al., 2021a). Riding this unprecedented wave, researchers reformulate the old wisdom by incorporating LLMs into neuro-symbolic reasoning, bypassing the inflexibility and ineffectiveness of domain-specific language-to-program parsers. Specifically, VisProg (Gupta & Kembhavi, 2022) pre-defines a set of visual modules, and uses LLMs to transform language instructions into symbolic programs consisting of the pre-defined visual modules. Taking a step forward, ViperGPT (Surís et al., 2023) releases the burden on manually-defined visual modules by introducing a code generator that could produce a code snippet based on each input instance of a new task.

Promising as these LLM-based neuro-symbolic models can be, they inevitably bear several weaknesses compared to the learning and reasoning processes of human beings. First, both VisProg and ViperGPT exhaustively produce one code snippet for each new instance of a task, which is extremely ineffective. This is in stark contrast with the human learning process: from an early age, we organically accumulate knowledge from particular experiences. Such knowledge acquired from specific cases could be reused and reconfigured, enabling us to quickly adapt to new tasks and new demands (Harlow, 1949; Mitchell et al., 1986; Lake et al., 2016; Ellis et al., 2023). The knowledge blocks grow progressively over time, gradually into a library with extraordinary richness and flexibility for fast generalization to any unseen task - the knowledge library that these models fall short of. Second, both models do not verify and examine the codes they generate. It seems that when the models generate a bad code snippet that cannot solve the input case, they just "let it go" without taking another stab for larger chance towards success. And of course, when they encounter similar cases again, they keep "stepping on the same rake". Human beings, on the other hand, would verify and examine the acquired knowledge by proposing a set of test scenarios before storing them in the library (Brulé & Blount, 1989). It's crucial that a neuro-symbolic reasoning model is equipped with the same abilities to verify the codes it produces, stores them in a library if satisfactory, and makes another attempt when the codes fail.

To this end, we introduce a novel Generative Neuro-symbolic Visual Reasoning Model (GENOME), proficient in assimilating new neural modules from a limited set of training examples. This model excels in handling standard visual reasoning tasks such as visual question answering. Addition-

ally, it demonstrates outstanding module transfer capabilities for tasks like image editing, and exhibits an exceptional ability to generalize to new reasoning tasks with limited training examples. As illustrated in Figure 2, GENOME comprises three stages: 1) module initialization, 2) module generation, and 3) module execution. From the initial training set examples, an LLM discerns the necessity for a new module to tackle the task and, if required, produces the respective input and output. In the second stage, LLMs implement and refine the new module, ensuring seamless integration with existing modules and resulting in accurate responses to the training queries. During testing, the LLM first converts language instructions into executable high-level programs like *COMPARE_ATTRIBUTE(IMAGE,BOX0,BOX1,ATTR)* for comparing attributes of different bounding boxes. The program will be run with the new module sets, producing the desired outputs.

We assessed the performance of GENOME across six visual reasoning tasks, spanning from visual question answering (Hudson & Manning, 2019) to Raven's Progressive Matrices (Zhang et al., 2019a). The experimental findings reveal that GENOME delivers competitive results on standard benchmarks, ensuring both transparency and interoperability. Notably, modules honed on these standard tasks can be adeptly adapted to diverse domains, including image editing and knowledge tagging (Gupta & Kembhavi, 2022). Additionally, with minimal training examples, GENOME demonstrates the capability to manage new visual reasoning tasks (Burke, 1985; Jiang et al., 2023a) by repurposing modules.

## 2 RELATED WORK

**Visual Reasoning.** Our work aims to handle visual reasoning tasks, which require a model to draw new inferences based on the acquired visual cues in images or videos (Hudson & Manning, 2019; Kazemzadeh et al., 2014; Goyal et al., 2017; Zhang et al., 2019a; Jiang et al., 2023a). Typical tasks for visual reasoning include visible question answering (Goyal et al., 2017; Hudson & Manning, 2019), visual grounding (Kazemzadeh et al., 2014; Yu et al., 2016; Chen et al., 2020) and Raven's Progressive Matrices (Burke, 1985; Zhang et al., 2019a). Various models (Hudson & Manning, 2018; Yu et al., 2018; Zhang et al., 2021; Ding et al., 2023) have been developed to handle these tasks but most of them are ad-hoc and carefully designed for a specific task, leaving it an open research question on how to build a general model that can handle different kinds of visual reasoning problems by only showing a few examples.

**Neuro-symbolic Visual Reasoning.** Our work is also closely related to neuro-symbolic visual reasoning models (Andreas et al., 2016b; Mao et al., 2019a; Chen et al., 2021c; 2022), where the models decompose the query of the visual reasoning tasks into a series of reasoning steps and represent each reasoning step with a neural module (i.e., a code snippet for achieving specific functions like localizing objects and recognizing object categories). While these models have better model interoperability and data efficiency than previous connectionist models (Hudson & Manning, 2018; Anderson et al., 2018), they often show their limitations in representing natural language instructions in the wild with the limited pre-defined reasoning steps (Yang et al., 2020; Chen et al., 2021b). Moreover, they need to manually define and implement each neural module one by one, making it hard to scale up and handle multiple tasks within a single model.

**Foundation Models for Reasoning.** Recently, large language models (LLMs) (Brown et al., 2020; Ouyang et al., 2022) have been widely used in language understanding (Hendrycks et al., 2020) and reasoning (Cobbe et al., 2021; Amini et al., 2019). Schick et al. (2023) develop the toolformer to show that LLMs can use external tools to better handle language tasks. Cai et al. (2023) shows that LLMs can make simple tools for natural language tasks by writing code snippets. LLMs have also been used in vision-language tasks. Most of these works (Li et al., 2023; Alayrac et al., 2022) connect LLMs with additional vision encoders and fine-tune them with massive vision-language pairs. As evaluated by Xu et al. (2023b), while these models show great performance on in-domain tasks, they also perform poorly on tasks out of the training domains. They are also extremely computation-expensive. For example, it takes 15 days to use 1536 TPUv4 for training Flamingo (Alayrac et al., 2022).

**LLMs for Programming.** There are some works that use LLMs to write codes to handle tasks. Pereira & Hartmann used LLMs to progressively enhance and specify system subcomponents, empowering users to develop versatile programs through a systematic iterative disambiguation method. Jiang et al. (2023b) learned to generate code with LLMs, which involves a planning phase for outlining solution steps and an implementation phase for generating code. Besides the dense engagement

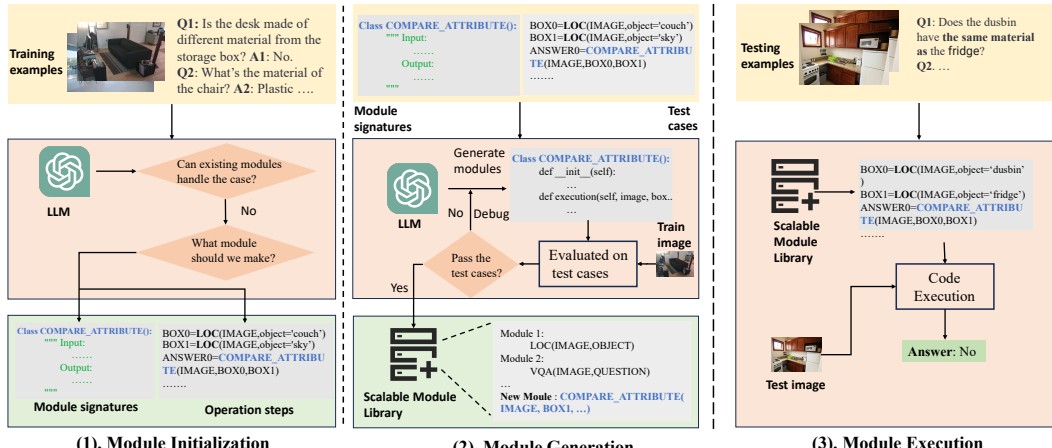

Figure 2: The framework of our GENOME, which contains three stages, module initialization, module generation, and module execution. In stage 1, we feed the questions and the signature of the existing modules to the LLM and ask it to identify whether we can handle the query within the operations of the existing modules. If not, we ask the LLM to generate the signature of the new module (i.e. the input and output) and predict the reasoning steps to handle the query task. In stage 2, we feed the module signature and the testing cases to the LLM and ask the LLM to implement the module and test its pass rate on the training examples. We only accept the modules that successfully handle the query. In stage 3, we first use the LLM to parse the query into symbolic operations and then execute these operations on the test images with the help of the scalable module library. We take VQA as an example and such a framework can also be expanded to other tasks like referring expression comprehension and Raven.

with the visual modalities as input, our GENOME differs from them in modularizing code snippets for better module expansion and reuse. These unique differences make our GENOME model have new capabilities like growing new modules to handle the visual reasoning tasks and transferring the modules into new domains. Some research works (Vendrow et al., 2023; Gao et al., 2023) also used LLMs and few-shot examples to improve AI models' performance. However, they only focused on improving the pure neural network model's performance and automatically discovered groups to use them for model design.

**Visual Programming by LLMs.** Another line of research has been combining vision models (Li* et al., 2022; Kirillov et al., 2023; Radford et al., 2021) with LLMs in an off-shelf manner. Early models (Yang et al., 2022; Chen et al., 2023b) transformed images into captions and append the captions into the LLMs' prompt to handle vision-language tasks. While the method is simple, they also perform inferior and lack transparency. Recently, VisPROG (Gupta & Kembhavi, 2022) uses LLMs to transform language instructions into pre-defined modular operations for step-by-step reasoning. However, it still requires manually implementing each module one by one. Later, ViperGPT (Surís et al., 2023) shows that the LLMs can be used to write a code snippet for each query instance independently to handle vision tasks. However, the code it writes has not been examined and tested by any training examples and there is no guarantee about the performance and code safety. Instead, we propose GENOME that ask LLMs to create new neural models (i.e. general code snippets to achieve specific functions) and handle the given tasks through only a few training examples. Our GENOME has the reliance on a few training examples to learn new modules. However, such newly generated modules can cooperate with each other and be reused for other tasks for better performance. There is also research like (Rahaman et al., 2021) which adopts pure neural network architecture to abstract the reasoning problem by generating the script and dynamically executing it. Differently, our GENOME is a neuro-symbolic method that provides better model transparency in explicit Python scripts and is able to make use of existing large pre-trained models to make and reuse new modules.

## 3 METHOD

### 3.1 OVERALL

In this section, we present a novel framework named as Generative Neuro-symbolic Visual Reasoning Model (GENOME) for the acquisition of neural modules and solutions of visual reasoning tasks with only a limited set of training examples. GENOME comprises several pre-defined opera-

tors that serve as the initial building blocks. Each neural operator corresponds to a neural module and is implemented using a Python code snippet, thereby enabling specific functions such as object localization within an image. Nevertheless, it is not possible to pre-define all the necessary neural modules prior to addressing the visual reasoning tasks. Consequently, there arises a need to generate new modules based on a limited number of visual reasoning task examples.

Figure 2 illustrates that GENOME consists of three distinct stages: 1) module initialization, 2) module generation, and 3) module execution. In the module initialization stage, when provided with a small set of training examples from the visual reasoning dataset, the primary objective is to determine whether the existing neural modules are sufficient to address the query examples. If the existing modules are inadequate for the query instance, GENOME will identify the requirements for creating new modules, including defining the modules' input and output specifications. During the module generation stage, GENOME leverages the LLM to implement the neural module based on the provided training examples and the specified input and output format (i.e., function signature) and add the module to the module library only when it passes the test cases. Once the new module is successfully implemented, the module execution orchestrates the transformation of input queries into a sequence of neural operations. Subsequently, these operations are applied to the neural modules to obtain the correct output. All these three stages are powered by robust code generation capabilities and the in-context learning technique. Prompts for each stage can be found at Figure 15-17

## 3.2 MODEL DETAILS

Utilizing a limited number of examples from the training set of visual reasoning tasks, we employ the GENOME framework, comprising three distinctive stages: module initialization, module generation, and module execution.

**Module Initialization.** The initial phase within our GENOME framework is module initialization, dedicated to determining the set of new modules required to address the visual reasoning task. As depicted in Figure 2-1, we employ an LLM to assess the feasibility of handling these training instances using existing neural modules. Should this not be achievable, we task the LLM with specifying the necessary new modules (e.g. *COMPARE_ATTRIBUTE* in Figure 2) for an accurate response to the query. The outcome of this stage comprises function signatures detailing the input and output formats for these new modules. Furthermore, it facilitates the transformation of the input query into a sequence of reasoning steps, which function as test cases to validate the correctness of the generated program within module generation. The prompt for the LLM in this stage is shown in Figure 15.

**Module Generation.** The second phase of our GENOME framework is module generation, which focuses on implementing the correct new modules proposed during the module initialization stage. Specifically, after receiving the signature of a new module, we incorporate all the corresponding test cases that call the new module into the prompt and employ learning-in-context techniques to generate multiple program candidates. Note that a new module is usually paired with multiple test cases. These program candidates are subsequently executed using the provided training examples. If a program encounters errors during execution, we incorporate the error information into the LLM's prompt and instruct it to rectify these issues. We only accept program candidates that achieve a pass rate surpassing a predefined threshold ($\eta$). This procedure bears resemblance to the code translation of LLMs discussed in (Chen et al., 2023a), but we extend it to accommodate more intricate multi-modal input types and instructions from natural language and raw images. The inclusion of module generation in the context of visual reasoning tasks offers two principal advantages. Firstly, it upholds the transparency and interpretability of neuro-symbolic models while preserving competitive performance. Secondly, it exhibits generative capabilities and scalability as our GENOME can autonomously generate new modules tailored to specific tasks.

**Module Execution.** Given the integration of newly-generated modules with existing neural modules tailored for visual reasoning, the GENOME framework initiates query parsing from the testing dataset, transforming them into executable operations through in-context learning. An illustrative prompt for this stage is depicted in Figure 17. Notably, although various visual reasoning tasks may possess distinct inputs and outputs, they can re-purpose these intermediary modules designed for other tasks to enhance overall performance. This feature represents a unique capability for code generation at the module level, an aspect hitherto unexplored by prior methods(Surís et al., 2023; Gupta & Kembhavi, 2022).

## 4 EXPERIMENTS

In this section, we present a comprehensive series of experiments to evaluate the performance of our models. Initially, we demonstrate our models' effectiveness in learning neural modules on two established benchmarks: GQA (Hudson & Manning, 2019), focusing on compositional visual question answering, and RefCOCO (Kazemzadeh et al., 2014), which assesses referring expression comprehension. Subsequently, we illustrate how the modules acquired from these two datasets can be successfully applied to novel tasks such as image editing and knowledge tagging. Moreover, we highlight the adaptability of our framework to address novel visual reasoning tasks (Raven (Zhang et al., 2019a) and MEWL (Jiang et al., 2023a)), even with limited training examples. Before delving into these experiments, we provide an overview of the experimental settings.

**Experimental Details.** The success of our GENOME relies on a set of pre-defined modules and APIs as the starting point. We utilize handcrafted modules from VisProg (Gupta & Kembhavi, 2022) as our initial components. Additionally, we incorporate several new APIs from ViperGPT to enhance module creation. We also include some new APIs from ViperGPT (Surís et al., 2023) for making new modules. In Section 4.4, we also include results parsed by the open-source LLM from WLM (Xu et al., 2023a) to investigate the influence of different LLM models. A comprehensive list of the pretrained modules employed in our approach can be found in Section A.1 of the Appendix. We extract training examples to acquire new modules. More precisely, we extracted 300 examples from GQA, 100 from RefCOCO, 10 from Raven, and 10 from MEWL.

**Datasets and Evaluation Metric.** We show experiments of our GENOME on standard vision-language benchmarks, GQA (Hudson & Manning, 2019) and RefCOCO Kazemzadeh et al. (2014). GQA is a popular compositional visual reasoning dataset with synthesis multi-hop questions, making it suitable for multi-step reasoning. RefCOCO is a typical visual grounding dataset, evaluating models' ability to localize objects and understand fine-grained spatial and semantic relationships. Following ViperGPT, we evaluate GQA on test-dev split and RefCOCO on the testA split. Then, we show GENOME's abilities on the other transferred tasks, image editing, and knowledge tagging and compare it with VisProg. Since the image editing and knowledge tagging datasets from VisProg are not publicly available, we built two new datasets for evaluation. The new editing dataset contains 50 images and instruction pairs. The new knowledge tagging dataset contains 50 images with 50 referring expressions. We provide more details about the dataset in Appendix A.4. The datasets will be released for research purposes. Finally, we show that GENOME can learn to handle new visual reasoning tasks like Raven (Zhang et al., 2019a) and MEWL (Jiang et al., 2023a) by observing a few training examples and module learning. Raven is a task for relational and analogical visual reasoning of image sets and has been widely used for non-verbal intelligence tests. MEWL is a recent benchmark proposed to assess how machines learn word meaning in grounded visual scenes. Examples of these tasks can be found at Figure 5 and Figure 6.

### 4.1 COMPARISON WITH BASELINES ON VISUAL REASONING.

We conducted analysis between our model and several baseline models using the GQA and RefCOCO datasets. Due to the deprecation of the original professional Codex API (`code-davinci-002`), we replaced it with the currently available API (`gpt-3.5-turbo-instruct`) and conducted experiments with both our model and the baseline models to ensure a fair comparison. We did not carry out experiments with GPT-4 due to the prohibitive cost.

The results, as presented in Table 1, demonstrate that our model achieves competitive performance in both visual question answering and referring expression comprehension, thus confirming its effectiveness. Furthermore, we provide an illustrative module from our model in Figure 11. This newly created module has the capability to utilize various available APIs to select attributes from the images. The step-by-step reasoning process of our model is detailed in Figure 3, offering greater transparency compared to end-to-end models.

| Methods | GQA | RefCOCO |
|---|---|---|
| BLIP-2 | 44.7 | - |
| KOSMOS-2 | - | 57.4 |
| ViperGPT-CodeX | 48.1 | 72.0 |
| VisPROG-Instruct | 45.4 | - |
| ViperGPT-Instruct | 38.2 | 62.4 |
| Ours-Instruct | 45.6 | 69.2 |

Table 1: Evaluation on standard visual reasoning benchmarks, GQA and RefCOCO.

### 4.2 GENOME FOR TRANSFER LEARNING

In this section, we demonstrate our model's robust capabilities in transfer learning. We augment the modular library by incorporating modules created

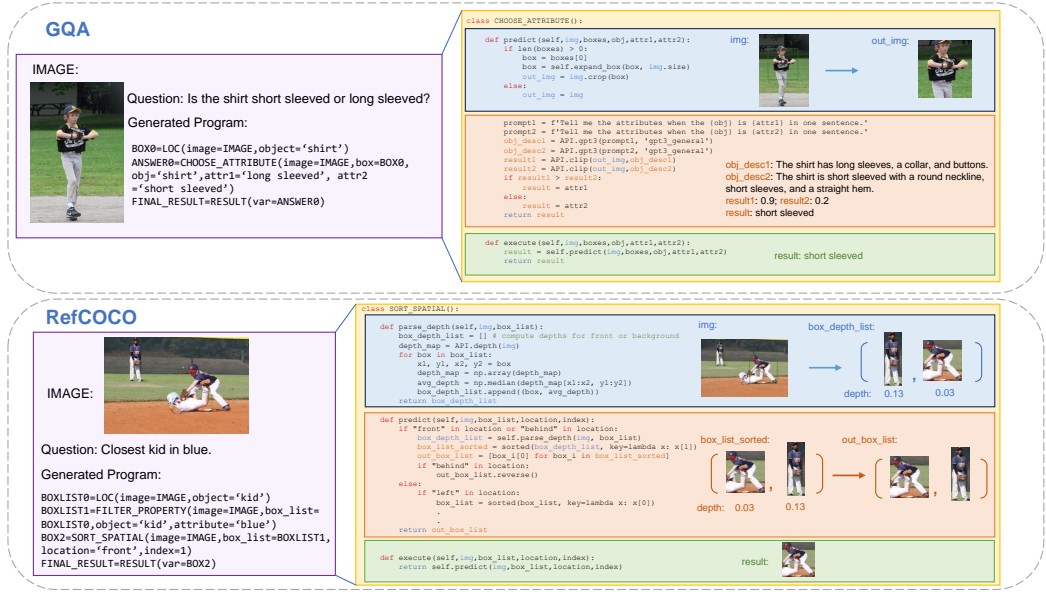

Figure 3: Qualitative examples of GENOME's on GQA and RefCOCO. The query images, language instructions, and the parsed programs are shown on the left. The corresponding new modules and the value of important variables are shown on the right.

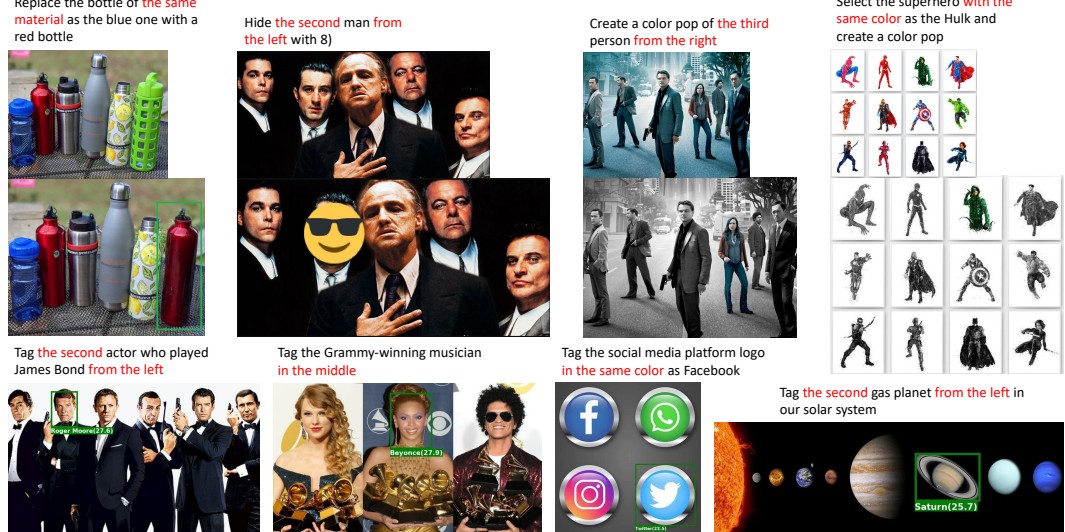

Figure 4: Qualitative examples of GENOME's on the image editing and knowledge tagging tasks. The language instructions of the tasks are shown the original images while the modified results of the images are shown below the original images. The emphasis of the instructions is highlighted with red colors, which requires our new modules to handle. While the key regions of the output images are bounded with green colors.

from GQA and RefCOCO, employing in-context examples to guide the Language Model (LLM) in generating step-by-step instructions for task execution. Qualitative results for this task are depicted in Figure 4. As illustrated in Figure 4, our model excels in generating semantically accurate images using the newly added module, whereas the baseline VisProg struggles to capture the required relationships with its fixed, pre-defined module library. To provide a more comprehensive evaluation of image editing, we enlist annotators to manually assess the correctness of the generated images. The models' performance is compared in Table 2, where our model outperforms the baseline. In the context of knowledge tagging, we task annotators with marking image regions referenced by ex-

| Methods | Image Editing | Tagging | | | Localization | | |
|---|---|---|---|---|---|---|---|
| | Accuracy | Precision | Recall | F1 | Precision | Recall | F1 |
| VisProg | 16.7 | 18.4 | 21.7 | 19.9 | 32.8 | 35.3 | 34.0 |
| GENOME | 55.3 | 67.1 | 52.3 | 58.8 | 76.9 | 57.9 | 66.0 |

Table 2: Evaluation of GENOME on Transfer Learning with image editing and knowledge tagging tasks. Our GENOME shows much better performance on all criteria, showing the effectiveness of the transferred modules. A qualitative comparison can be seen Figure 7 in the Appendix.

| Methods | Center | L-R | U-D |
|---|---|---|---|
| ResNet+DRT | 58.1 | 65.8 | 67.1 |
| ALANS-V | 98.4 | 97.3 | 96.4 |
| GENOME | 80.1 | 67.6 | 69.1 |
| Human | 95.5 | 86.4 | 81.8 |

| Methods | shape | color | material |
|---|---|---|---|
| Aloe | 34.2 | 33.2 | 31.0 |
| Flamingo-1.1B | 49.3 | 35.3 | 48.5 |
| GENOME | 43.7 | 45.3 | 41.0 |
| Human | 92.4 | 87.2 | 72.7 |

Table 3: Evaluation of GENOME on Raven (Zhang et al., 2019a). Compared with methods trained with massive in-domain data, our model performs competitively.

Table 4: Evaluation of GENOME on MEWL (Jiang et al., 2023a). Compared with approaches trained on extensive in-domain data, our model shows competitive performance.

pressions and employ the same metrics as RefCOCO for evaluating the accuracy of bounding boxes and employ the BERT score to assess the correctness of labeled names. Our model demonstrates superior performance in both image editing and knowledge tagging. We show a typical example in Figure 7 of the Appendix to show how our GENOME make use of new modules to perform better knowledge tagging result the baseline.

### 4.3 GENOME ON FEW-SHOT TASK LEARNING.

As a general module learning framework, our model is not only able to learn new modules to handle existing tasks but also can learn to handle new visual reasoning tasks from a few training examples. We evaluate such abilities on new tasks, Raven (Zhang et al., 2019a) and MEWL (Jiang et al., 2023a). Specifically, we first prompt the LLM to learn pattern recognition modules for visual understanding and then ask the LLM to generate a solver module to handle the task. The instances of our model prediction are shown in Figure 5 and Figure 6. Note that visual reasoning from Raven is widely used in intelligent testing for humans, which shows our model's strong capabilities and potential. We report the performance of our model and the baselines in Table 3 and Table 4. Our model is significantly better than previous fully-supervised methods like ResNet+DRT (Zhang et al., 2019a) and Aloe (Ding et al., 2021), showing its effectiveness. Note that all these models ResNet+DST, ALANS-V (Zhang et al., 2022), Aloe (Ding et al., 2021) and Flamingo (Alayrac et al., 2022) are models fully-finetuned on in-domain data, while our GENOME is a general few-shot framework to learn modules for problem-solving. Moreover, we can observe the new compositionality and module re-usage from Figure 8 of the Appendix. Although the SOLVER module was originally learned from center-type problems, it can be naturally transferred to other types like left-right and up-down.

### 4.4 ABLATIONS

To gauge the efficacy of our model, we conducted a series of ablation studies addressing the following key inquiries: **Q1** How effective is module learning? **Q2** What impact does the quantity of training examples have on model performance? **Q3** How crucial is the LLM's capability for optimal performance? In our experiments, *GENOME w/o ML* represents a configuration without any new module learn-

| Methods | RefCOCO |
|---|---|
| GENOME w/o ML | 62.3 |
| GENOME-WLM | 64.4 |
| GENOME (10) | 49.4 |
| GENOME (50) | 67.0 |
| GENOME (100) | 67.1 |

Table 5: Ablation study of GENOME on RefCOCO.

ing but relies heavily on ViperGPT and VisProg-defined modules, directing the LLM to pinpoint a region matching the referring expression. On the other hand, *GENOME-WLM* replaces the `gpt-3.5-turbo-instruct` API with `WizardCoder-Python-34B-V1.0` from WizardLM (Xu et al., 2023a). The designations *GENOME (10)/(50)/(100)* indicate models trained with

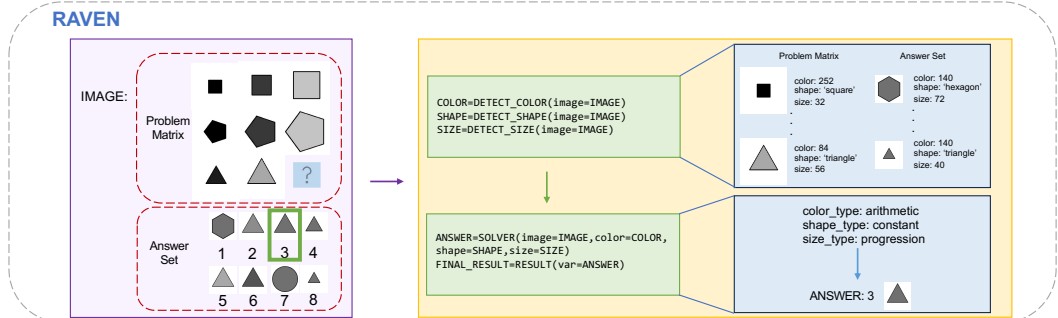

Figure 5: A qualitative example from the Raven dataset (Zhang et al., 2019a) is provided. This task involves a set of images with varying visual attributes, such as colors, shapes, and locations. Models are tasked with identifying the image that best matches the missing item in the Problem Matrix. GENOME exhibits the capability to compose modules (i.e. DETECT_SHAPE and SOLVER) for detecting these attribute rules and constructing a solver module to address the task. The correct answer is indicated by a green box.

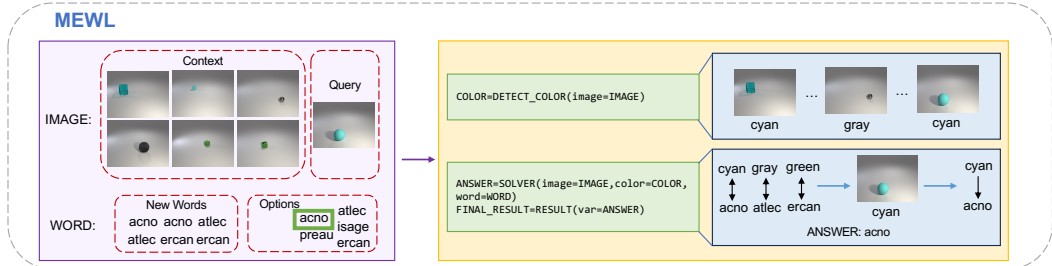

Figure 6: A qualitative illustration from the MEWL dataset (Jiang et al., 2023a) is presented. This task entails a set of images featuring diverse visual attributes, such as material and shapes, and it necessitates models to determine the word that corresponds to the query image. GENOME demonstrates the capability to generate modules for identifying these attribute rules and composing a solver module to address the task. The correct answer is indicated by a green box.

10, 50, and 100 examples, respectively. For resource constraints, we limited our experimentation to 800 RefCOCO samples.

Table 5 presents the outcomes, leading to these insights: module learning, given sufficient test instances, can bolster task performance (**Q1** addressed). A paucity of training examples, such as 10 for RefCOCO, might induce overfitting, but this diminishes with increased training data (50 examples), improving overall performance (**Q2** addressed). Finally, model performance appears intrinsically tied to the LLM's capacity, with superior LLMs delivering enhanced results (**Q3** addressed).

## 5 CONCLUSION

In this study, we introduce GENOME, which is designed to tackle visual reasoning tasks when confronted with limited training data. This approach combines language models to parse natural language into executable operations and create specialized visual modules tailored to the given task. Our model exhibits competitive performance on conventional tasks, effortless transfer of acquired modules to novel tasks, and the capability to adapt to new tasks even with limited training data. Our GENOME also proposes numerous avenues for future research. Firstly, it still necessitates task-specific prompts for each distinct reasoning task, and it would be intriguing to explore the use of a universal prompt for all tasks. Secondly, the framework can be extended to encompass a broader range of multi-modal reasoning tasks, incorporating diverse inputs such as audio, video, and tactile information.

## 6 ACKNOWLEDGEMENT

This work was supported by DSO grant DSOCO21072. We would also like to thank the computation support from AiMOS, a server cluster for the IBM Research AI Hardware Center.

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

# A    APPENDIX

In this section, we substantiate our claims in the paper by providing additional implementation details (Section A.1), more experimental analysis (Section A.2), exemplar prompts for each stage (Section A.3), details on dataset collection (Section A.4), qualitative examples of new learned modules (Section A.5).

## A.1    IMPLEMENTATION DETAILS

**Pre-defined Modules and API models.**    The success of our model still requires a set of pre-defined APIs. Following the modules in VisProg and ViperGPT, we adopt the following APIs. We adopt GLIP (Li* et al., 2022) for object localization. We adopt *gpt-3.5-turbo-instruct* from OpenAI and *WizardCoder-Python-34B-V1.0* from WizardLM for code generation. We use BLIP (Li et al., 2023) for answering simple questions about images. We use CLIP (Radford et al., 2021) and X-VLM (Zeng et al., 2021) for image-text classification. We use MiDaS (Ranftl et al., 2021) for estimating depths in images. We use stable diffusion (Rombach et al., 2022) for modifying image patches. Based on these APIs, we construct a set of pre-defined modules following VisProg. These pre-defined modules will be used to cooperate with the new learned modules for visual reasoning.

| Descriptions | Modules |
|---|---|
| Image Understanding | **Loc** for object location, **FaceDet** for face detection, **Select** and **Filter_Property** for image-text classification. **Filter_Spatial** for selecting image regions; |
| Image Manipulation | **Replace** for image editing, **colorPop** for changing images colors, **BgBlur** for blurring background, **Tag** for annotating box regions and **Emoji** for face tagging. **Crop** and its variants for cropping patches from the images. |
| Others | **List** for retrieving factual knowledge, **Count** for counting object numbers, **Eval**, **Result**, **BOX2MASK** and **MASK2BOX** for formatting outputs. |

Table 6: Pre-defined Modules used in GENOME.

**Details on Raven and MEWL.**    Note that a visual reasoning task does not necessarily use language as input. All we need is to prompt the LLMs to generate modules that recognize the patterns and solve the problem. In RAVEN, by prompting LLM, we can obtain DETECT_COLOR, DETECT_SHAPE, and DETECT_SIZE. The image is fed into these modules and the output is the color, shape, and size of the image. In this way, the input image is converted into a (color, shape, size) triplet. We provide LLM with ten examples from the RAVEN train split to demonstrate how to deduce the pattern of these triplets. By observing few-shot demonstrations, we let LLM generate the SOLVER() module, which detects the pattern of input triplets from the Problem Matrix and chooses the most appropriate answer from the Answer Set. Therefore, the internal of the SOLVER() module is primarily based on judgment, used to identify the patterns of the input triples in the Problem Matrix, thereby finding the answer in the Answer Set. The workflow of RAVEN is shown in Figure 5. As for MEWL, we employ a similar approach to handle it. One example is provided Figure 6. Since MEWL and RAVEN have different patterns, the SOLVER() module is not shared between RAVEN and MEWL. Thus, it utilizes distinct logic.

## A.2    MORE EXPERIMENTAL ANALYSIS.

**Computational Efficiency of Module Reuse.**    Our modularized design and module reuse strategy offer higher computational efficiency and generate shorter code compared to baseline approaches like ViperGPT, which creates solutions from scratch without modular abstraction and reuse. We calculated the average token count for both our GENOME model and ViperGPT, which lacks a module reuse mechanism, in their interactions with LLMs. As shown in table 7, our GENOME model's solutions are demonstrably shorter and more efficient.

| Methods | GQA | RefCOCO |
|---------|-----|---------|
| ViperGPT | 153.7 | 109.1 |
| GENOME | 62.3 | 54.4 |

Table 7: Comparison of average token number for the generated solutions on GQA and RefCOCO.

**More Ablation Study on Different Components.** To dissect different components of our proposed method, we showcase a more comprehensive and detailed ablation study here. We randomly select 800 samples from GQA test-dev split to further investigate the effectiveness of different components of GENOME. Moreover, we add baseline experiments of RAVEN and MEWL for better comparison. To better present all experimental results, all the ablation studies are organized into the following sections.

| Methods | GQA |
|---------|-----|
| GENOME | 45.9 |
| GENOME w/o input and output format | 43.2 |
| GENOME w/o good initialization | 41.8 |
| GENOME w/o existing modules in prompt for module making | 45.0 |
| GENOME w/o creating new modules | 44.7 |
| GENOME w/ random sampling | 44.3 |
| GENOME (60) | 44.5 |
| GENOME (120) | 45.3 |
| GENOME (300) | 45.9 |
| GENOME w/ different LLM | 44.3 |
| GENOME w/o debugging | 44.9 |

Table 8: More ablation on GQA dataset.

**Ablation on Prompt Design.** We conducted a series of experiments to observe the impact of prompt design on the overall performance of GENOME. Firstly, we removed the descriptions of input and output formats from the prompt. After removing these descriptions, the performance of GENOME dropped by 2.7%. This is because, without clear guidance on input and output formats, the modules might output in the wrong format, leading to errors in subsequent parsing of the results. Furthermore, on top of removing the input and output format, we also removed some of the in-context examples and descriptions about module signatures from the prompt. The performance further declined. Since our method consists of three stages: module initialization, module generation, and module execution, where module initialization is the first step of our method. Without adequate module initialization as a foundation, the subsequent results are largely impacted. Therefore, we can see that without good initialization, our performance drops by 4.1%.

Regarding the use of existing modules and creating new ones, from Table 8, we can observe that not using the predefined modules from VisProg results in a 0.9% decrease in our performance. This demonstrates the robust module generation capability of GENOME. Even without a series of predefined modules, our method can still build modules from scratch, solve problems, and the performance does not drop significantly. If we don't create new modules, then we are merely using the predefined modules. We can see that the result is 44.7%, which is 1.2% lower than our result of 45.9%. This performance gap highlights the effectiveness of the newly generated modules. By generating and using new modules, we can achieve better results.

**Ablation on Sampling.** In this section, we introduce our sampling strategy at first. Then, we conduct an experiment to showcase how the sampling methods will impact GENOME performance. Subsequently, we investigate how the number of training samples affects our results of different tasks.

**Sampling Strategy.** Our sampling strategy: the GQA dataset contains five structural types: choose, logical, compare, verify, and query. These structural types inspired the idea of generating our new modules. Taking COMPARE_COLOR as an example, this newly generated module is generated to address questions related to color within the compare structural type. From the visualization

| # of Samples | RAVEN | | | MEWL | | |
|---|---|---|---|---|---|---|
| | Center | L-R | U-D | shape | color | material |
| 5 | 46.5 | 37.2 | 39.8 | 38.9 | 39.6 | 37.9 |
| 10 | 80.1 | 67.6 | 69.1 | 43.7 | 45.3 | 41.0 |
| 20 | 80.1 | 67.6 | 69.1 | 43.7 | 45.3 | 41.0 |

Table 9: Number of Sampling Examples on RAVEN and MEWL.

of GQA, it is apparent that the query type can be addressed using the existing VQA module from VisProg, and problems in the logical type can be decomposed into sub-problems of choose, compare, and verify types. Therefore, when selecting training samples, we randomly chose 100 samples each from the choose, compare, and verify types. Altogether, these three types comprise 300 samples, all sourced from the GQA train split. Hence, we are not cherry-picking our training samples; rather, we are selecting training samples based on the structural types of GQA.

To explore the impact of sampling strategies on our experiment, we conducted an additional experiment with a random sampling of 300 samples, beyond our initial sampling strategy. In this setting, we randomly sampled 300 examples from the GQA train split. The performance was observed to be 44.3%, a decrease of 1.6% compared to 45.9%. This result suggests that a strategic sampling method can more effectively guide the LLM in generating more efficient modules for a given task. Relatively speaking, our method is robust in the face of choices in sampling strategies.

**Number of Sampling Examples.** We conduct a series of experiments to illustrate how the number of training samples influences the performance.

In the GQA and RefCOCO datasets, if a small number of training samples are used, it's possible for the generated modules to overfit certain samples, thereby reducing the generalization capability of the newly generated modules. Such overfitting in new modules can negatively impact the final results. Therefore, we can observe that when the number of samples is small, the performance of GENOME is poorer. As the number of samples increases, the effectiveness of GENOME improves. However, with a further increase in the number of samples, the performance gains of GENOME tend to saturate.

Regarding RAVEN and MEWL, since their patterns of change are limited, the number of few-shot samples selected is sufficient if it already covers all the variation patterns in RAVEN and MEWL. In other words, if the number of samples exceeds this threshold, there won't be any further improvement in the results; if it's below this threshold, the performance will decline. We selected 10 few-shot samples each in RAVEN and MEWL. As can be seen from the results in the table above, if the number of samples is equal to 5, there is a noticeable decrease in performance. This is because 5 few-shot samples are not enough to cover all the variation patterns of RAVEN or MEWL. If the number of samples is equal to 10 or 20, at this point, the few-shot samples are sufficient to encompass all possible variations. In this case, the same results are obtained.

**Ablation on LLM's capability.** By using a better LLM, our prompts can be better understood, and the LLM will generate higher-quality modules. In this experiment, we compared the results of using `gpt-3.5-turbo-instruct` (i.e., GENOME) and `gpt-3.5-turbo` (i.e., different LLM). Our experimental results show that better outcomes are achieved when using the more effective `gpt-3.5-turbo-instruct`. It is evident that the capabilities of the LLM influence the performance of GENOME. As the abilities of LLMs continue to improve, so will the performance of GENOME. Thanks to the flexibility of GENOME, once a better LLM is available, we can easily switch to the latest LLM to achieve better results.

**Ablation on Debug Mechanism.** The error-correction prompt contains the error message from Python interpreter and wrong code snippet. We prompt the LLM to correct the wrong code based on the error message from Python interpreter. We heuristically set the maximal number of debug iterations as 5. If the wrong code can be corrected within 5 iterations, we will keep it. Otherwise, it will be abandoned. (Details can be found in the Module Generation section of Figure 2) The errors mainly stem from two sources: one is basic syntax errors in Python code, such as indentation and variable name errors. The other source is some fundamental logical errors, such as mistakes made when setting variable types, like treating a variable that should be of the bool type as the string

| Methods | RAVEN | | | MEWL | | |
|---------|--------|------|------|-------|-------|----------|
| | Center | L-R | U-D | shape | color | material |
| VisProg variant | 36.8 | 26.1 | 27.8 | 35.2 | 35.9 | 34.9 |
| ViperGPT variant | 40.6 | 30.7 | 32.4 | 37.8 | 38.2 | 36.7 |
| Ours | 80.1 | 67.6 | 69.1 | 43.7 | 45.3 | 41.0 |

Table 10: Compare our GENOME model with baselines, VisProg, and ViperGPT on RAVEN and MEWL.

type. By observing the table above, we can conclude that the debug process can assist GENOME with generating more useful modules to elevate performance and prevent elementary programming mistakes.

**Additional Baseline Experiments of RAVEN and MEWL.** For RAVEN and MEWL, we have implemented the ViperGPT and VisProg baseline experiments in the following way. For VisProg, it requires a manual implementation of all modules by making use of the provided APIs. Thus, to enable VisProg to handle Raven and MEWL tasks, we manually implement and debug new hand-crafted modules for VisProg to recognize and discover patterns to handle the task. We call this baseline VisProg variant. We also put the training examples in GENOME' stage 1 into the prompt of VisProg variant for better performance. For ViperGPT, it has no manual modules and ask the LLMs to make use of the APIs to handle the instances. Thus, we manually write solutions for the training examples into the prompt of the ViperGPT to teach ViperGPT to handle the task. We call this approach ViperGPT variant. We have added such analysis into the revised paper. VisProg by itself needs a handcrafted solver module to find the target solution and it would be extremely difficult for ViperGPT to generate a solver from scratch. Thus, we add the solver module learnt from our GENOME model to pre-defined API pool of VisProg and ViperGPT. As shown in table 10, our GENOME model achieves better performance than these two baselines, showing the great value of module learning for handling new tasks from only a few examples.

| New Modules | GQA |
|-------------|-----|
| VERIFY_ATTRIBUTE | 14.1 |
| CHOOSE_ATTRIBUTE | 10.8 |
| VERIFY_COLOR | 6.9 |
| COMPARE_ATTRIBUTE | 5.9 |
| VERIFY_MATERIAL | 3.6 |

Table 11: Percentage of top-5 most-used new modules in GQA.

**More Details about New Modules.** We further take GQA as an example to exhibit the percentage of top-5 most-used new modules in GQA in the Table 11. The data in Table 11 shows the proportion of the five most common new modules appearing in the generated high-level programs. Overall, 38.7% of all generated high-level programs use the newly generated modules (this 38.7% calculation includes other less common modules and excludes duplicate samples, such as a single high-level program containing multiple new modules). From these results, it can be seen that these newly learned modules can be widely applied to GQA, thereby helping GENOME achieve good results on GQA.

**Additional Experiment on I-RAVEN.** As independently demonstrated in (Hu et al., 2021) and (Spratley et al., 2020), the Raven dataset (Zhang et al., 2019a) exhibits flaws in its choice design, enabling models to learn shortcuts for solving the RPM reasoning task. Thus, we additionally conduct the balanced I-RAVEN dataset (Hu et al., 2021) for further analysis. Following our setting in RAVEN (Zhang et al., 2019a), we use 10 training samples for learning new modules to handle the task and test the model on the testing set. As shown in table 12, our GENOME is still able to handle the abstract reasoning task with high accuracy and data efficiency.

| Methods | Center | L-R | U-D |
|---|---|---|---|
| LEN (Zheng et al., 2019) | 56.4 | 44.2 | 44.2 |
| CoPINet (Zhang et al., 2019b) | 54.4 | 51.9 | 52.5 |
| SRAN (Hu et al., 2021) | 78.2 | 70.1 | 70.3 |
| GENOME | 85.2 | 74.6 | 75.4 |

Table 12: Experiments for Raven's Progressive Matrices on I-RAVEN dataset (Hu et al., 2021).

**Effectiveness of Module Learning.** To better investigate the effectiveness of our GENOME, we further two variants of our GENOME to show the effectiveness of module learning. **GENOME w/o ML** represents a configuration without any new module learning but relies heavily on ViperGPT and VisProg-defined modules, directing the LLM to answer the question related to the image content and pinpoint a region matching the referring expression. It strictly follows the function call style like VisProg and our GENOME. We also develop a variant of **GENOME w/o ML v2** that allows the LLM to call functions like ViperGPT with standard control flow and arbitrary Python logic. For resource constraints, we limited our experimentation to the randomly selected 800 examples from GQA and RefCOCO with `gpt-3.5-turbo-instruct` API. As shown in table 13, our GENOME performs better than both two baselines across tasks, showing the effectiveness of the module learning.

| Methods | GQA | RefCOCO |
|---|---|---|
| GENOME w/o ML | 43.3 | 62.3 |
| GENOME w/o ML v2 | 40.9 | 65.5 |
| GENOME | 45.9 | 67.1 |

Table 13: Ablation of module learning on GQA and RefCOCO.

### A.3 PROMPTS FOR EACH STAGE.

The ability of Our GENOME is from in-context learning of LLMs (Brown et al., 2020), when the prompts are keys to tell what the LLM should generate. We show the exemplar prompts of our models to learn the VQA task in Figure 15-17.

### A.4 DETAILS AND EXAMPLES OF THE NEW DATASETS.

To evaluate Knowledge Tagging, 50 tagging instructions are annotated on 50 internet images including personalities and a variety of objects such as logos, flowers, buildings, fruits and sports, among others. For each instruction, we manually annotated the ground truth bounding box and the associated tag. For image editing assessment, we collected 50 editing instructions on 50 images including personalities and various objects like foods, furniture, animals, utensils etc. 25 images are from the COCO dataset and the other 25 images are from the internet. For the image editing tasks, we ask three annotators to estimate whether the editing is correct or not. For the knowledge tagging task, we consider the localization is correct if the detected region has an IoU higher 0.5 with the ground-truth annotation. For text tagging, we compare the prediction with the annotated text with BERT matching (BEM) (Bulian et al., 2022). If the matching score is higher than 0.5, we consider it a successful matching. More examples of the two datasets can be found at Figure 9 and Figure 10.

### A.5 QUALITATIVE EXAMPLES.

In this subsection, we show the qualitative examples of the learned modules and qualitative cases of how they handle different tasks. We show an example of GENOME performs better than VisProg in Figure 7. At the top of Figure 7, our model effectively utilizes the COMPARE_COLOR module acquired from GQA to pinpoint the correct region, whereas VisProg fails to generate the correct program due to its rigid module library. Figure 8 highlights emerging forms of compositionality and module re-usage. Notably, although the *SOLVER* module was originally trained on center-type

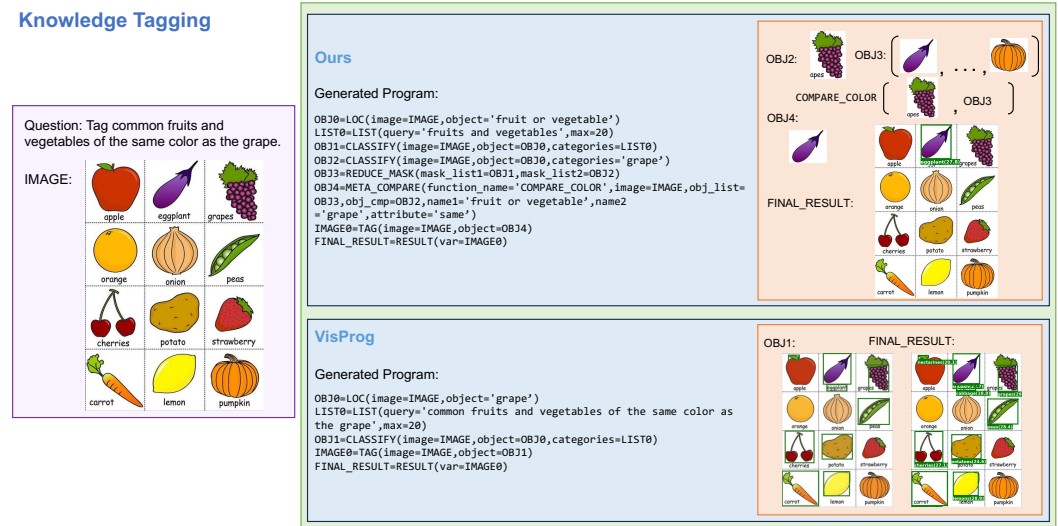

Figure 7: A typical example of how our GENOME outperforms VisProg on knowledge tagging. In the top, our model is able to make use of the *COMPARE_COLOR* module learned from GQA to localize the correct region while VisProg fail to generate the correct program with its fixed module library.

problems within the Raven dataset, it demonstrates inherent adaptability to other problem types, including left-right and up-down orientations.

**New Learned Modules.** We show the examplar new learned modules from the GQA and Ref-COCO in Figure 11-14. As shown in Figure 11, the new learned module ( CHOOSE_ATTRIBUTE) is able to use the LLM to retrieve relevant knowledge first and then adopt the image-text classifier to match the attributes. In Figure 13-14, we see that the new module SORT_SPATIAL is able to localize objects with spatial index.

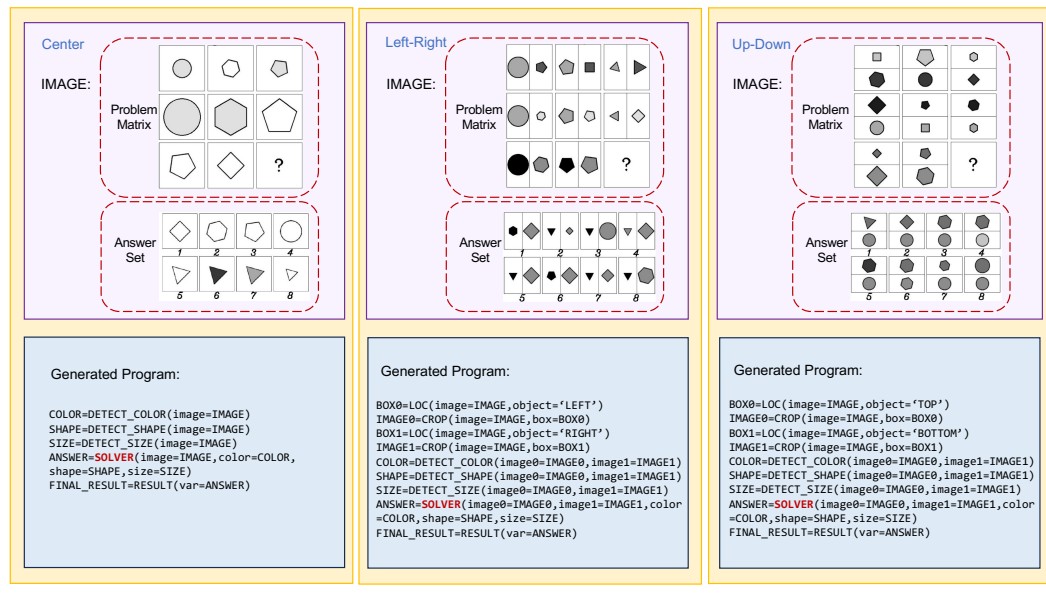

Figure 8: New compositionality and module re-usage in the Raven dataset. While the SOLVER module was initially trained on center-type problems in the Raven dataset, it exhibits a natural transferability to other types, such as left-right and up-down problems.

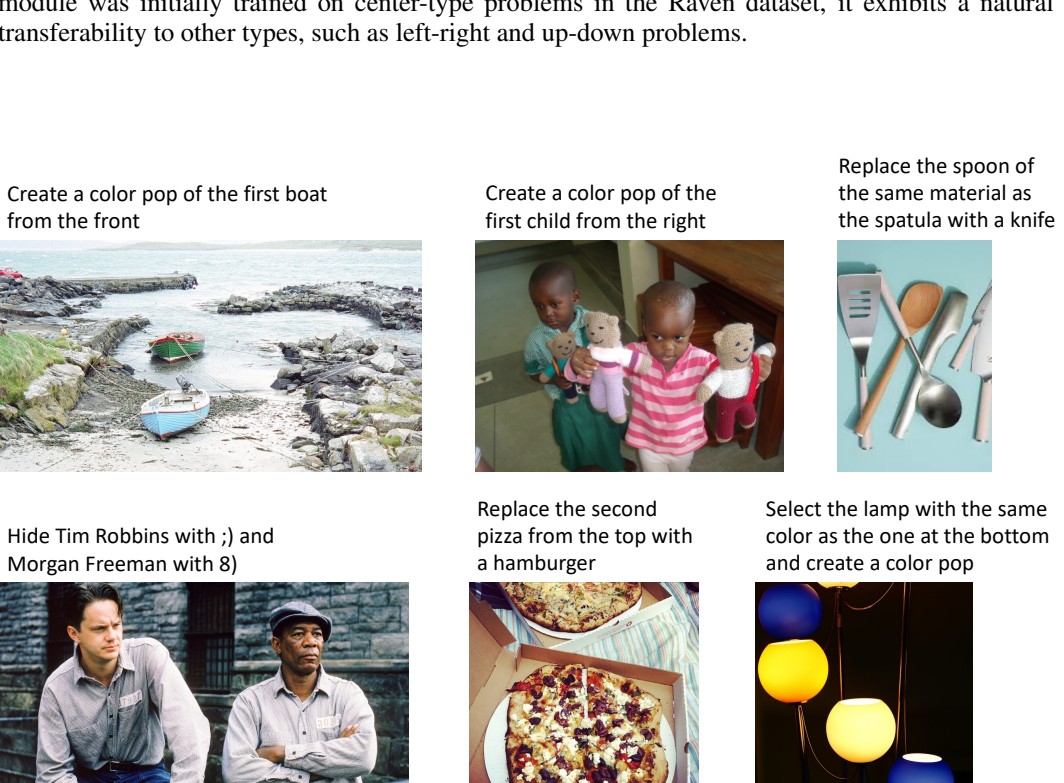

Figure 9: More examples of the new image edit dataset. The dataset asks models to edit images' fine-grained and regional details according to diverse language instructions.

Tag the famous landmark of Europe in the bottom right

Tag the common bird of the same color as the ibis

Tag the second famous film director from the left

Tag the famous painting of the Louvre in the left

Tag the second dog from the right

Tag the second Nobel Laureate in Physics from the left

Figure 10: More examples of the new knowledge tagging dataset. The dataset requires models to localize the target region and tag the region with the desired information.

```python
class CHOOSE_ATTRIBUTE():
    """
    Input:
    image: an image object
    box: a list of bounding boxes
    object: a string
    attribute1: a string
    attribute2: a string
Output:
    result: a string
    Examples:
    Question: Is the coat thick or thin?
    BOX0=LOC(image=IMAGE,object='coat')
    ANSWER0=CHOOSE_ATTRIBUTE(image=IMAGE,box=BOX0,object='coat',
    attribute1='thick',attribute2='thin')
    FINAL_RESULT=RESULT(var=ANSWER0)
    """
    step_name = 'CHOOSE_ATTRIBUTE'

    def __init__(self):
        print(f'Registering {self.step_name} step')

    def expand_box(self,box,img_size,factor=1.5):
        W,H = img_size
        x1,y1,x2,y2 = box
        dw = int(factor*(x2-x1)/2)
        dh = int(factor*(y2-y1)/2)
        cx = int((x1 + x2) / 2)
        cy = int((y1 + y2) / 2)
        x1 = max(0,cx - dw)
        x2 = min(cx + dw,W)
        y1 = max(0,cy - dh)
        y2 = min(cy + dh,H)
        return [x1,y1,x2,y2]

    def predict(self,img,boxes,obj,attr1,attr2):
        if len(boxes) > 0:
            box = boxes[0]
            box = self.expand_box(box, img.size)
            out_img = img.crop(box)
        else:
            out_img = img
        prompt1 = f'Tell me the attributes when the {obj} is {attr1} in
        one sentence.'
        prompt2 = f'Tell me the attributes when the {obj} is {attr2} in
        one sentence.'
        obj_desc1 = API.gpt3(prompt1, 'gpt3_general')
        obj_desc2 = API.gpt3(prompt2, 'gpt3_general')
        result1 = API.clip(out_img,obj_desc1)
        result2 = API.clip(out_img,obj_desc2)
        if result1 > result2:
            result = attr1
        else:
            result = attr2
        return result

    def execute(self,img,boxes,obj,attr1,attr2):
        result = self.predict(img,boxes,obj,attr1,attr2)
        return result
```

Figure 11: Exemplar generated module from the GQA dataset. This automatically constructed module can make use of different APIs to compare attributes of an image region.

```python
class COMPARE_COLOR():
    """
    Input:
        image: an image object
        box1: a list of bounding boxes
        box2: a list of bounding boxes
        object1: a string
        object2: a string
        compare_type: a string
    Output:
        result: a string
    """
    def expand_box(self,box,img_size,factor=1.5):
        W,H = img_size
        x1,y1,x2,y2 = box
        dw = int(factor*(x2-x1)/2)
        dh = int(factor*(y2-y1)/2)
        cx = int((x1 + x2) / 2)
        cy = int((y1 + y2) / 2)
        x1 = max(0,cx - dw)
        x2 = min(cx + dw,W)
        y1 = max(0,cy - dh)
        y2 = min(cy + dh,H)
        return [x1,y1,x2,y2]
    def predict(self,img,boxes1,boxes2,obj1,obj2,compare_type):
        if len(boxes1) > 0:
            box1 = boxes1[0]
            box1 = self.expand_box(box1,img.size)
            out_img1 = img.crop(box1)
        else:
            out_img1 = img
        if len(boxes2) > 0:
            box2 = boxes2[0]
            box2 = self.expand_box(box2,img.size)
            out_img2 = img.crop(box2)
        else:
            out_img2 = img
        color1 = API.vqa(out_img1, f'What color is the {obj1}?')
        color2 = API.vqa(out_img2, f'What color is the {obj2}?')
        prompt = f'Can the {color1} be regarded as the same color as'
        f'{color2}? You should just reply yes or no without any other
        words.'
        temp = API.gpt3(prompt, 'gpt3_general')
        if 'same' == compare_type:
            if 'yes' in temp.lower():
                result = 'yes'
            elif 'no' in temp.lower():
                result = 'no'
        elif 'different' == compare_type:
            if 'yes' in temp.lower():
                result = 'no'
            elif 'no' in temp.lower():
                result = 'yes'
        else:
            if 'yes' in temp.lower():
                result = 'yes'
            elif 'no' in temp.lower():
                result = 'no'
        return result
    def execute(self,img,boxes1,boxes2,obj1,obj2,compare_type):
        result = self.predict(img,boxes1,boxes2,obj1,obj2,compare_type)
        return result
```

Figure 12: Exemplar generated module from the GQA dataset.

```python
class SORT_SPATIAL():
    """
    Select objects from the image that match the spatial location.
    Objects are represented by the bounding boxes.
    Returns the bounding boxes that satisfie the condition.
    Input:
        image: raw PIL image
        box_list: a list of unnormalized bounding boxes
        location: the location can only be left, middle, right, top,
        bottom, front and behind
        index: a number for the rank the object
    Output:
        box: a bounding box
    Examples:
        Question: second sandwich from the right on the bottom
        BOXLIST0=LOC(image=IMAGE,object='sandwich')
        BOXLIST1=SORT_SPATIAL(image=IMAGE,box_list=BOXLIST0,location=
        'right',index=2)
        BOXLIST2=SORT_SPATIAL(image=IMAGE,box_list=BOXLIST1,location=
        'bottom',index=1)
        FINAL_RESULT=RESULT(var=BOXLIST2)
    """
    step_name = 'SORT_SPATIAL'
    def predict(self,img,box_list,location,index):
        if index < 0 or index > len(box_list):
            return []
        if index == 0:
            return [box_list[0]]
        if "front" in location or "behind" in location:
            box_depth_list = self.parse_depth(img, box_list)
            box_list_sorted = sorted(box_depth_list, key=lambda x: x[1])
            out_box_list = [box_i[0] for box_i in box_list_sorted]
            if "behind" in location:
                out_box_list.reverse()
        else:
            if "left" in location:
                box_list = sorted(box_list, key=lambda x: x[0])
            elif "right" in location:
                box_list = sorted(box_list, key=lambda x: x[2], reverse
                    =True)
            elif "top" in location:
                box_list = sorted(box_list, key=lambda x: x[1])
            elif "bottom" in location:
                box_list = sorted(box_list, key=lambda x: x[3], reverse
                    =True)
            else:
                return []
            if index > len(box_list):
                return []
            out_box_list = [box_list[index-1]]
        return out_box_list
    def check_location(self,img,box,location):
        w, h = img.size
        x1, y1, x2, y2 = box
        cx = (x1 + x2) / 2
        cy = (y1 + y2) / 2
        if 'left' in location:
            if cx > w / 2:
                return False
```

Figure 13: Exemplar generated module from the RefCOCO dataset. The rest part of the code is in Figure 14.

```
1           elif 'right' in location:
2               if cx < w / 2:
3                   return False
4           if 'top' in location:
5               if cy > h / 2:
6                   return False
7           elif 'bottom' in location:
8               if cy < h / 2:
9                   return False
10          return True
11
12      def parse_depth(self,img,box_list):
13          box_depth_list = []
14          # compute depths for front or background
15          depth_map = API.depth(img)
16          for box in box_list:
17              x1, y1, x2, y2 = box
18              depth_map = np.array(depth_map)
19              avg_depth = np.median(depth_map[x1:x2, y1:y2])
20              box_depth_list.append((box, avg_depth))
21          return box_depth_list
22
23      def execute(self,img,box_list,location,index):
24          return self.predict(img,box_list,location,index)
```

Figure 14: Exemplar generated module from the RefCOCO dataset. The former part of the code is in Figure 13. This generated module is able to localize objects based on their location in images and the depth of images.

```
1  Pre-defined Modules:
2  class LOC():
3      """
4      Generate boxes of the object on the image.
5      Input:
6          image: an image object
7          object: an object string
8      Output:
9          box: a list of bounding boxes
10         Examples:
11             BOX0=LOC(image=IMAGE,object='camel')
12         """
13 class COUNT():
14     """
15     Count the number of boxes in the list.
16     Input:
17         box: a list of bounding boxes
18     Output:
19         number: number of boxes
20     Examples:
21         ANSWER0=COUNT(box=BOX1)
22     """
23 Suppose you are a program expert. Given a set of pre-defined modules,
24 could you identify whether it is possible to write a program to get the
25 answer to the question? If not, what new modules do we need?
26 Note that you can only use the below pre-defined modules:
27 LOC, COUNT, CROP .......
28
29 Question: Is the purse to the left or to the right of the person?
30 Yes. The program is:
31 BOX0=LOC(image=IMAGE,object='person')
32 IMAGE0=CROP_LEFTOF(image=IMAGE,box=BOX0)
33 BOX1=LOC(image=IMAGE0,object='purse')
34 ANSWER0=COUNT(box=BOX1)
35 ANSWER1=EVAL(expr=f"'left' if {ANSWER0} > 0 else 'right'")
36 FINAL_RESULT=RESULT(var=ANSWER1)
37
38 Question: Which object is larger, the sphere or the blue cube?
39 No. We need to make a new module "COMPARE_SIZE" first. Here is the header
40 of the class:
41 class COMPARE_SIZE():
42     """
43     Compare the size of two objects in the image.
44     One object is identified by the first bounding box of box0
45     Another object is identified by the first bounding box of box1
46     Input:
47         image: an image object
48         box0: a list of bounding boxes
49         box1: a list of bounding boxes
50     Output:
51         flag: return True if first object is larger else False
52     Examples:
53         Question: Which object is larger, the sphere or the blue cube?
54         BOX0=LOC(image=IMAGE,object='sphere')
55         BOX1=LOC(image=IMAGE,object='blue cube')
56         FLAG0=COMPARE_SIZE(image=IMAGE,box0=BOX0,box1=BOX1)
57         ANSWER2=EVAL(expr=f"'sphere' if {FLAG0} else 'blue cube'")
58         FINAL_RESULT=RESULT(var=ANSWER)
59     """
60 .......
61 Question: __INSERT_NEW_QUESTION__
```

Figure 15: Prompt of module initialization (Stage 1) to propose the module candidates.

```
1   Pre-defined APIs:
2   class API():
3       def loc(cls, image: Image.Image, obj_name: str) -> List[List[int]]:
4           """
5           Returns boxes of the specific object on the image.
6           Input:
7               image: an image object
8               obj_name: a text string
9           Output:
10              boxes: a list of bounding boxes
11          Example:
12              [[261, 160, 525, 299]] = API.loc(image, "camel")
13          """
14          return boxes
15      def gpt3(cls, prompt: str) -> str:
16          """
17          Returns response from gpt3
18          Input:
19              prompt: prompt to gpt3
20          Output:
21              response: gpt3 response
22          Example:
23              "joe biden" = API.gpt3("Who is the US President?")
24          """
25          return response
26      .......
27
28  Given the above APIs, could you write a new module, Loc()?
29  class Loc():
30      """
31      Generate boxes of the object on the image.
32      Input:
33          img: an image object
34          obj_name: an object string
35      Output:
36          selected_boxes: a list of bounding boxes
37      Examples:
38          BOX1=LOC(image=IMAGE0,object='food')
39      """
40  Response:
41  class Loc():
42      step_name = 'LOC'
43      def execute(self,img,obj_name):
44          if obj_name=='TOP':
45              bboxes = [self.top_box(img)]
46          elif obj_name=='BOTTOM':
47              bboxes = [self.bottom_box(img)]
48          elif obj_name=='LEFT':
49              bboxes = [self.left_box(img)]
50          elif obj_name=='RIGHT':
51              bboxes = [self.right_box(img)]
52          else:
53              bboxes = self.predict(img,obj_name)
54          return bboxes
55
56      def predict(self,img,obj_name):
57          return API.loc(img,obj_name)
58      .......
59  Given the above APIs, could you write a new module, __MODULE_NAME__?
60  __MODULE_HEAD__
```

Figure 16: Prompt of module generation (Stage 2) to make a module based on the module's input and output.

```
1   Think step by step to answer the question.
2
3   You can only use modules below:
4   LOC
5   COUNT
6   EVAL
7   RESULT
8   VERIFY_ATTRIBUTE
9   VERIFY_COLOR
10  VERIFY_MATERIAL
11  .......
12
13  Question: Is the vehicle in the top of the image?
14  Program:
15  BOX0=LOC(image=IMAGE,object='TOP')
16  IMAGE0=CROP(image=IMAGE,box=BOX0)
17  BOX1=LOC(image=IMAGE0,object='vehicle')
18  ANSWER0=COUNT(box=BOX1)
19  ANSWER1=EVAL(expr=f"'yes' if {ANSWER0} > 0 else 'no'")
20  FINAL_RESULT=RESULT(var=ANSWER1)
21
22  Question: Who is carrying the umbrella?
23  Program:
24  BOX0=LOC(image=IMAGE,object='umbrella')
25  IMAGE0=CROP(image=IMAGE,box=BOX0)
26  ANSWER0=VQA(image=IMAGE0,question='Who is carrying the umbrella?')
27  FINAL_RESULT=RESULT(var=ANSWER0)
28
29  Question: Do the towel and the box have a different colors?
30  Program:
31  BOX0=LOC(image=IMAGE,object='towel')
32  BOX1=LOC(image=IMAGE,object='box')
33  ANSWER0=COMPARE_ATTRIBUTE(image=IMAGE,box1=BOX0,box2=BOX1,object1='towel'
34  ,object2='box',attribute='color',question=QUESTION)
35  FINAL_RESULT=RESULT(var=ANSWER0)
36
37  Question: Is the knife made of ceramic?
38  Program:
39  BOX0=LOC(image=IMAGE,object='knife')
40  ANSWER0=VERIFY_MATERIAL(image=IMAGE,box=BOX0,material='ceramic',object=
41  'knife',question=QUESTION)
42  ANSWER1=EVAL(expr=f"'yes' if {ANSWER0} else 'no'")
43  FINAL_RESULT=RESULT(var=ANSWER1)
44
45  Question: Is the coat thick or thin?
46  Program:
47  BOX0=LOC(image=IMAGE,object='coat')
48  ANSWER0=CHOOSE_ATTRIBUTE(image=IMAGE,box=BOX0,object='coat',attribute1=
49  'thick',attribute2='thin')
50  FINAL_RESULT=RESULT(var=ANSWER0)
51  .......
52
53  Question: __INSERT_NEW_QUESTION__
54  Program:
```

Figure 17: Prompt of module execution (Stage 3) to parse programs for a new test case.

