# OpenReview forum: "GENOME: Generative Neuro-Symbolic Visual Reasoning by Growing and Reusing Modules"
_ICLR.cc/2024/Conference — ICLR 2024 poster_

### Official Review · Reviewer_JpcA · 2023-10-26

**Soundness:** 3 good
**Presentation:** 3 good
**Contribution:** 3 good
**Rating:** 6
**Confidence:** 4

**Summary:**

The authors tackle the problem of prior visual programming methods exhaustively generating entire code snippets for each new task. Instead, they proposed GNSVR to grow and re-use a library of modules. GNSVR first initializes new modules based on a train set, then evaluates the new modules based on additional few-shot training examples, and finally it executes the modules on the full test set. It shows comparable performance on visual reasoning tasks, can transfer modules to new tasks, and adapt to new visual reasoning tasks with few training examples.

**Strengths:**

I am appreciative of the idea of generating more modular and composable modules that are verified, to be used in a library of skills for future tasks. I think the idea of continuously growing this library is innovative, and the method simple and elegant.

**Weaknesses:**

W1. Is there anything to prevent overfitting to the small train set of a given task, and only creating specific modules tailored for that domain? I can imagine that these overfitted modules may not be very useful in new tasks.

W2. Isn’t it possible that the method creates a broad function signature, but during stage 2 verification with the small set of train examples, it overfits to a bad implementation that only works for those examples, and therefore actually harms performance from there on out? I’m mainly concerned about the above two overfitting challenges.

W3. There seems to be an assumption that the queries in the train set actually all require similar modules, for example, to verify the new modules, the selected set of test cases from the train set must actually use those modules. I’m not sure if this is a reasonable assumption, and also, how is this sampling of train examples (both in stage 1 and stage 2) done?

W4. In the experiments, are Visprog/Viper also given the same few-shot training set? For example, I believe you can use the same method of error correction and correct both Visprog/Viper when it is incorrect. In this way, you can include Visprog/Viper comparison in Tables 3 and 4. Would be great to better disentangle what drives this improvement of performance -- modularity in the proposed modules, or the training examples given in the loop, etc.

**Questions:**

Q1. Please clarify the selection of training examples used in stage 1 and 2.

Q2.  “If a program encounters errors during execution, we incorporate the error information into the LLM’s prompt and instruct it to rectify these issues.” How is this done?

---

> ### Author Response · Authors · 2023-11-17
> **Look forward for your  response (Part 1)**
>
> Thank you for the positive comments and insightful suggestions.
>
> > Q1. Is there anything to prevent overfitting to the small train set of a given task, and only creating specific modules tailored for that domain? I can imagine that these overfitted modules may not be very useful in new tasks.
>
> >Q2. Isn’t it possible that the method creates a broad function signature, but during stage 2 verification with the small set of train examples, it overfits to a bad implementation that only works for those examples, and therefore actually harms performance from there on out? I’m mainly concerned about the above two overfitting challenges.
>
> **Q1-Q2. About overfitting VS training number.**
> Thanks for concerns on the overfitting issue of train examples. Our GNSVR pipeline do have overfitting issue when the number of the training examples is too small (10), which could be reflected in the ablation study in table 5 of the revised paper. However, as the number of training examples goes up (50 and 100), the performance becomes stable, which is unlikely that the GNSVR framework creates a broad function signature overfitting the training examples. We believe that one reason for the strong few-shot performance of GNSVR is that all our pre-defined APIs are general APIs that work well for vision tasks in the wild. We also provide the performance VS test cases in the following tables for different tasks. As shown in the tables, as the training number is larger than a small number, our GNSVR framework becomes effective and relieve the overfitting issue.
>
> **Number of Sampling Examples**
>
> Table 1: Number of Sampling Examples on GQA.
> | # of samples | GQA |
> | ------- | -------- |
> | 60     | 44.5   |
> | 120    | 45.3   |
> | 300    | 45.9   |
>
> Table 2: Number of Sampling Examples on RefCOCO.
> | # of samples | RefCOCO |
> | ------- | -------- |
> | 10     | 49.4   |
> | 50     | 67.0   |
> | 100    | 67.1   |
>
> Table 3: Number of Sampling Examples on RAVEN.
> | # of samples | Center | L-R  | U-D  |
> | ------------ | ------ | ---- | ---- |
> | 5     | 46.5   | 37.2 | 39.8 |
> | 10    | 80.1   | 67.6 | 69.1 |
> | 20    | 80.1   | 67.6 | 69.1 |
>
> Table 4: Number of Sampling Examples on MEWL.
> | # of samples | shape | color |  material  |
> | ------------ | -------- | -------- | --- |
> | 5     | 38.9   | 39.6 | 37.9 |
> | 10    | 43.7   | 45.3 | 41.0 |
> | 20    | 43.7   | 45.3 | 41.0 |

---

> > ### Author Response · Authors · 2023-11-17
> > **Look forward for your response (Part 2)**
> >
> > >Q3. There seems to be an assumption that the queries in the train set actually all require similar modules, for example, to verify the new modules, the selected set of test cases from the train set must actually use those modules. I’m not sure if this is a reasonable assumption, and also, how is this sampling of train examples (both in stage 1 and stage 2) done?
> >
> > The training samples in stage 1 and stage 2 are the same samples, randomly sampled by question categories. Specially, we first sample N examples to perform stage 1 to get the new module signatures and a set of corresponding test cases. The module signatures contain the standard input and output formats of the new modules. A test case is generated by parsing the corresponding sample query into a high-level program. We consider a new generated module passes a test case if the execution of the high-level program generates the correct answer. We also group the training examples according to the same new module signature, and use them to generate a set of test cases that would be used to test the correctness of the new modules.
> >
> > >Q4. W4. In the experiments, are VisProg/ViperGPT also given the same few-shot training set? For example, I believe you can use the same method of error correction and correct both VisProg/ViperGPT when it is incorrect. In this way, you can include VisProg/ViperGPT comparison in Tables 3 and 4. Would be great to better disentangle what drives this improvement of performance -- modularity in the proposed modules, or the training examples given in the loop, etc.
> >
> > **Q4. Variants of VisProg and ViperGPT.** Thanks for your suggestion on implementing variants of VisProg and ViperGPT for Raven and MEWL. Note that both VisProg and ViperGPT have no training phase that **generates** new modules for **reusage** in the test phase. Thus, there is no easy way for VisProg and ViperGPT to perform error correction with test cases. During the rebuttal, we have implemented the new variants in the following way to make use of training examples for VisProg and ViperGPT.
> > For VisProg, it requires a manual implementation of all modules by making use of the provided APIs. Thus, to enable VisProg handle Raven and MEWL tasks, we manually implement and debug new hand-crafted modules for VisProg to recognize and discover patterns to handle the task. We call this baseline **VisProg variant**. We also put the training examples in GNSVR' stage 1 into the prompt of **VisProg variant** for better performance. For ViperGPT, it has no manual modules and ask the LLMs to make use of the APIs to handle the instances. Thus, we manually write solutions for the training examples into the prompt of the ViperGPT to teach ViperGPT to handle the task. We call this approach **ViperGPT variant**. We have added such analysis into the revised paper. VisProg by itself needs a handcrafted solver module to find the target solution and it would be extremely difficult for ViperGPT to generate a solver from scratch. Thus, we add the solver module learnt from our GNSVR model to pre-defined API pool of VisProg and ViperGPT. As shown in table 5 and table 6 below, our GNSVR model achieves better performance than these two baselines, showing the great value of module learning for handling new tasks from only a few examples.
> >
> > Table 5: Compare our GNSVR model with baselines, VisProg and ViperGPT on RAVEN.
> > | Methods | Center | L-R  | U-D  |
> > | ------- | ------ | ---- | ---- |
> > | VisProg variant  | 36.8   | 26.1 | 27.8 |
> > | ViperGPT variant | 40.6   | 30.7 | 32.4 |
> > | Ours    | 80.1   | 67.6 | 69.1 |
> >
> > Table 6: Compare our GNSVR model with baselines, VisProg and ViperGPT on MEWL.
> > | Methods | shape | color |  material  |
> > | ------- | -------- | -------- | --- |
> > | VisProg variant | 35.2   | 35.9 | 34.9 |
> > | ViperGPT variant | 37.8   | 38.2 | 36.7 |
> > | Ours    | 43.7   | 45.3 | 41.0 |

---

> > > ### Comment · Reviewer_JpcA · 2023-11-20
> > > **Response to authors**
> > >
> > > Thank you for the thoughtful response! Can you please elaborate on how the training examples are "randomly sampled by question categories"? I believe this relates to my concerns on overfitting, and how strategies are required (that may use additional information, like knowledge of question categories in a dataset) to ensure a broad coverage of modules are proposed. Thanks!

---

> ### Author Response · Authors · 2023-11-21
> **Thanks for your constructive comments.**
>
> Thanks again for your response on the overfitting concern of the GNSVR model. We further explain your concerns as followed. In GQA, we have sampled training examples by question categories. This strategy does require the usage of question categories in the training dataset. In refCOCO, RAVEN and MEWL, we sample examples randomly since there is no sub category by types.
>
> To furtehr investigate our model's ability against overfitting, we have conducted new experiments on Table A below. We randomly sample different training samples from the whole training set 3 times with different random seeds, which we denote it **Random Sampling**. We also perform another sampling strategy that randomly sample training examples by question types, which we call **Random Sampling by types**. Based on the results in table A, we can see that our model works in both **Random Sampling** and **Random Sampling by types**, achieving reasonable performance although we do observe that **Random Sampling** strategy has larger variances on GQA. We will add such analysis in the revised paper.
>
> Table A: Ablation on GQA for sampling strategies.
> | Method GNSVR              | Accuracy |
> | --------                  | -------- |
> |  Random Sampling          | 44.8 +- 0.41 |
> |  Random Sampling by types        | 45.9 +- 0.14 |
>
> Another way to show our GNSVR model's ability against **overfitting** is that as shown in Table 2, Fig. 4, and Fig. 7,  of the paper, **the learned modules learned from GQA and refCOCO can be generalized to new tasks like image editing and knowledge tagging.** Note that language instructions and images from image editing and knowledge tagging are quite different from that of the GQA's and RefCOCO's images.

---

### Official Review · Reviewer_vHZ5 · 2023-10-31

**Soundness:** 3 good
**Presentation:** 3 good
**Contribution:** 3 good
**Rating:** 6
**Confidence:** 4

**Summary:**

This paper proposes GNSVR, a neuro-symbolic system that grows a library of modules for visual reasoning tasks through the use of LLMs. It consumes a small set of example training tasks, and generates new modules with a two-step process. In step 1, a LLM is queried to see if an example task can be solved given functions from a base API, if the LLM responds negatively, then it is instructed to propose a new module to add into the API that would help solve this task, as well as to produce an I/O example for how this module would be used to solve the task. In step 2, given doc-string descriptions of these would-be useful modules, an LLM is prompted to author a python function, potentially referencing base-API logic, that matches the description and corresponds to its previously generated I/O example. At test-time, these modules are integrated into the API, and a LLM uses API calls to produce a program that can reason over visual input matching the logic of the input question. The system is primarily evaluated against other neuro-symbolic systems for visual reasoning that operate with a fixed API (VisProg, ViperGPT). Compared to the previous systems, GNSVR offers competitive performance on standard benchmarks (GQA, RefCOCO), and demonstrates promising results that its discovered modules can help generalize to related tasks, like image editing and knowledge tagging.

**Strengths:**

In general, I think the direction of the proposed method is sound and compelling; systems like VisProg and ViperGPT provide promising ways to push neuro-symbolic reasoning past toy-domains, but they are limited by fixed APIs. Using LLMs to augment these APIs in a task-specific manner is an interesting idea with many potential avenues for future exploration.

From a methodological stand-point, I think the division of labor between the module initialization step and the module generation step is a neat insight. This design decision allows the LLM to create it's own tests, that can then be used to help guarantee that LLM generated functions 'do what they are supposed to'.

The experimental results are largely positive for the proposed method, although I have some concerns about their design that I'll detail below. That said, I think the experimental results do support the claim that the proposed system offers improvements over both VisProg and ViperGPT across tasks. Perhaps the most compelling claim is that the modules that GNSVR finds from the GQA and RefCOCO tasks can "transfer" for related tasks of image editing and knowledge tagging, supported by the fact that GNSVR significantly outperforms VisProg in this setting. The evaluations against fully-supervised methods on the Raven and MEWL tasks is also fairly impressive, considering GNSVR is operating in a few-shot learning paradigm.

**Weaknesses:**

# Main Concern

From my perspective, the biggest current weakness of the paper is that from the experimental design its hard to parse out exactly how the discovered modules affect the system's performance. Ostensibly, this can be gleaned from comparisons between GNSVR and VisProg/ViperGPT, but there are more differences between these systems beyond merging in discovered modules. Specifically, GNSVR uses a "base API" that is a combination of VisProg and ViperGPT, so the "fair" comparison would be against an ablated version of GNSVR that removes steps 1 and 2, and just tries to solve test-problems with the original API functions. This condition is considered in the ablation experiment (GNSVR w/o ML), but only a subset of the RefCOCO test-set. To solidify the claim that the improvement GNSVR observes stems from its discovered modules, this base condition should be added to all of the experimental set-ups (tables 1-5), for example, from Table 2 its unclear how much of the delta improvement between VisProg and GNSVR can be attributed to improvements in the base API versus improvements to the API from the new modules.

Beyond this, I'm also slightly concerned about the design of the GNSVR w/o ML baseline. At inference time, is this baseline allowed to invoke arbitrary python logic in the style of ViperGPT (e.g. standard control flow constructs) or is it restricted to *only* using API function calls in the style of VisProg. I would imagine that the first condition would be more fair to evaluate GNSVR. Solving some tasks might require simple logic that the LLM knows how to express in python, but might not be directly expressible with a series of API calls. In GNSVR, this logic is incorporated into modules, but in the baseline the LLM should still have the opportunity to invoke similar logic in its test-time solutions (otherwise its impossible to properly evaluate the usefulness of the discovered modules). Please clarify which of these modes the baseline is operating in.

# Minor

Compared to VisProg and ViperGPT this system seems to require more training data, as the I/O pairs are not only used to populate in-context examples, but also impact (i) what module concepts are proposed and (ii) how the correctness of each module concept is evaluated. This point about reliance on training data is touched on in the ablation section, but it would be good to make this distinction explicit when comparing the pros/cons of the proposed system against past work.

**Questions:**

While the broad-stroke description of the method is clearly presented, many of the important details were left unspecified (making reproducibility challenging without a reference implementation). To improve clarity, and help understanding of the results, I think the below questions would be important to answer in future revisions:

# 1. Module Generation

(a) Can new modules be hierarchical (e.g. can one proposed module referenced a previously proposed module), or can they only call out to the original API functions?

(b) The error-correction and pass-rate logic for the module generation step are not clear. What is the "pass-rate" at which a function is accepted as a new module, from the listed examples it seems like a single input/output pair is used, so is the pass rate 1?

(c) What exactly is the error-correcting mechanism when one of the authored functions produces an error -- is it sent back through the LLM with a modified prompt? How many times? What constitutes an error? This seems like a potentially important part of the contribution, so I would encourage the authors to even consider adding an ablation condition demonstrating this is helpful for the performance of the system.

(d) It would be informative to provide additional details on the generated modules for each task: how many are created for each task? How often does each one get used in solving test-cases? Do they each capture distinct concepts, or are some modules duplicates?

# 2. In-context examples

I don't believe there is any information into how in-context examples are chosen, which can greatly affect LLM performance.

(a)  From the example prompts, it looks like a single function is given as an in-context example for step 1 and step 2, are these always COMPARE_SIZE and LOC, as shown in figures 15 and 16?

(b) For the inference in-context examples are these chosen randomly from the training tasks that found a successful program?

# 3. MISC

(a) For the Raven task, my understanding is that the input is just a series of images. If this understanding is correct, how do you turn these images into a question? I am also quite curious to see the internals of the SOLVER generated module, is this module shared between Raven and MEWL, or does it employ distinct logic?

---

> ### Author Response · Authors · 2023-11-17
> **Look forward for your constructive response (Part 1)**
>
> Thank you for the constructive comments and insightful suggestions.
>
> >Q1. From my perspective, the biggest current weakness of the paper is that from the experimental design its hard to parse out exactly how the discovered modules affect the system's performance. Ostensibly, this can be gleaned from comparisons between GNSVR and VisProg/ViperGPT, but there are more differences between these systems beyond merging in discovered modules. Specifically, GNSVR uses a "base API" that is a combination of VisProg and ViperGPT, so the "fair" comparison would be against an ablated version of GNSVR that removes steps 1 and 2, and just tries to solve test-problems with the original API functions. This condition is considered in the ablation experiment (GNSVR w/o ML), but only a subset of the RefCOCO test-set. To solidify the claim that the improvement GNSVR observes stems from its discovered modules, this base condition should be added to all of the experimental set-ups (tables 1-5), for example, from Table 2 its unclear how much of the delta improvement between VisProg and GNSVR can be attributed to improvements in the base API versus improvements to the API from the new modules.
>
> **Q1. More ablation study on Performance**
>
> Thanks for the insightful suggestions. We add more baseline experiments for comparison.
>
> For the GQA and RefCOCO baseline's comparison, please refer to **Q2** below.
>
> In Table 2 of the main paper, we showcase the capability of GNSVR's transfer learning. That is to say, the new modules learned from GQA and RefCOCO can be applied to the image editing and knowledge tagging task. Therefore, in the two tasks involved in Table 2, we did not learn any new modules but instead transferred and used new modules learned from other tasks. In Table 2, we can see that in all metrics, we surpassed VisProg. This performance improvement actually stems from the new modules generated from GQA and RefCOCO. It is these new modules that enabled functions that the inherent modules of VisProg couldn't achieve, thus leading to the improved performance of our system.
>
> As for RAVEN and MEWL, we have implemented the ViperGPT and VisProg baseline experiments in the following way.
> For VisProg, it requires a manual implementation of all modules by making use of the provided APIs. Thus, to enable VisProg handle Raven and MEWL tasks, we manually implement and debug new hand-crafted modules for VisProg to recognize and discover patterns to handle the task. We call this baseline **VisProg variant**. We also put the training examples in GNSVR' stage 1 into the prompt of **VisProg variant** for better performance. For ViperGPT, it has no manual modules and ask the LLMs to make use of the APIs to handle the instances. Thus, we manually write solutions for the training examples into the prompt of the ViperGPT to teach ViperGPT to handle the task. We call this approach **ViperGPT variant**. We have added such analysis into the revised paper. VisProg by itself needs a handcrafted solver module to find the target solution and it would be extremely difficult for ViperGPT to generate a solver from scratch. Thus, we add the solver module learnt from our GNSVR model to pre-defined API pool of VisProg and ViperGPT. As shown in table 1 and table 2, our GNSVR model achieves better performance than these two baselines, showing the great value of module learning for handling new tasks from only a few examples.
>
> Table 1: Compare our GNSVR model with baselines, VisProg and ViperGPT on RAVEN.
> | Methods | Center | L-R  | U-D  |
> | ------- | ------ | ---- | ---- |
> | VisProg variant  | 36.8   | 26.1 | 27.8 |
> | ViperGPT variant | 40.6   | 30.7 | 32.4 |
> | Ours    | 80.1   | 67.6 | 69.1 |
>
> Table 2: Compare our GNSVR model with baselines, VisProg and ViperGPT on MEWL.
> | Methods | shape | color |  material  |
> | ------- | -------- | -------- | --- |
> | VisProg variant | 35.2   | 35.9 | 34.9 |
> | ViperGPT variant | 37.8   | 38.2 | 36.7 |
> | Ours    | 43.7   | 45.3 | 41.0 |
>
> >Q2. Beyond this, I'm also slightly concerned about the design of the GNSVR w/o ML baseline. At inference time, is this baseline allowed to invoke arbitrary python logic in the style of ViperGPT (e.g. standard control flow constructs) or is it restricted to only using API function calls in the style of VisProg. I would imagine that the first condition would be more fair to evaluate GNSVR. Solving some tasks might require simple logic that the LLM knows how to express in python, but might not be directly expressible with a series of API calls. In GNSVR, this logic is incorporated into modules, but in the baseline the LLM should still have the opportunity to invoke similar logic in its test-time solutions (otherwise its impossible to properly evaluate the usefulness of the discovered modules). Please clarify which of these modes the baseline is operating in.

---

> > ### Author Response · Authors · 2023-11-17
> > **Look forward for your constructive response (Part 2)**
> >
> > **Q2. the detailed implementation of (GNSVR w/o ML baseline)**
> > Thanks for the concerns about the baseline's implementation. Our current implementation of the GNSVR w/o ML baseline is restricted to only using API function calls in the style of VisProg. To make it fair, we have wrapped the APIs in ViperGPT into **the same style** as VisProg and our GNSVR.
> > We highly value the reviewer's opinion and develop a new baseline variant, **GNSVR w/o ML v2**, where the LLM is allowed to invoke arbitrary python logic. s shown in table 3, our GNSVR performs better than both
> > two baselines across tasks, showing the effectiveness of the module learning.
> >
> > Table 3: Comparison of GNSVR and baselines on RefCOCO and GQA.
> > | Model               | GQA     | RefCOCO|
> > | -------- |-------- | -------- |
> > | **GNSVR w/o ML**    | 43.3    |  62.3  |
> > | **GNSVR w/o ML v2** | 40.9    |  65.5  |
> > | **GNSVR**           | 45.9    |  67.1  |
> >
> >
> >
> > >Q3. Compared to VisProg and ViperGPT this system seems to require more training data, as the I/O pairs are not only used to populate in-context examples, but also impact (i) what module concepts are proposed and (ii) how the correctness of each module concept is evaluated. This point about reliance on training data is touched on in the ablation section, but it would be good to make this distinction explicit when comparing the pros/cons of the proposed system against past work.
> >
> > **Q3. Discussion on the pros/cons of the proposed system against past work** Thanks for the suggestion to make the reliance on training examples distinct and explicit. Compared with existing methods, our GNSVR framework can leveraging LLMs to create new neural models or general code snippets for specific functions. Moreover, these newly generated modules in GNSVR can cooperate and be reused for various tasks, enhancing overall performance and adaptability. However, compared with existing frameworks VisProg and ViperGPT, we also need a few training examples to serve as test cases to learn new modules. We have also added the pros/cons of the proposed system in the related work section highlighted in the blue color in the revised version.
> >
> > > Q4. (a) Can new modules be hierarchical (e.g. can one proposed module referenced a previously proposed module), or can they only call out to the original API functions?
> >
> > **Q4. About hierarchical module generation.** We do **not** explicitly restrict the LLM to only use the original API functions. However, as we examine the generated modules one by one. We found that all the modules use the original APIs functions. The reasons are that we have provided examples in the prompt that make use of original APIs for new modules. However, we do believe that it is an interesting direction to include hierarchical module generation in the future work.
> >
> > > Q5. (b) The error-correction and pass-rate logic for the module generation step are not clear. What is the "pass-rate" at which a function is accepted as a new module, from the listed examples it seems like a single input/output pair is used, so is the pass rate 1?
> >
> > **Q5. More Details about Pass Rate Definition.** Since we have been using multiple training examples for our module initialization. Thus, each initialized new module should be paired with multiple test cases (*i.e.* multiple instances of the visual reasoning tasks) and we only accept the modules that whose pass rate is larger than a threshold (0.5 in our implementation). We have added such explanation for pass rate in the revised paper.
> >
> > > Q6. (c) What exactly is the error-correcting mechanism when one of the authored functions produces an error -- is it sent back through the LLM with a modified prompt? How many times? What constitutes an error? This seems like a potentially important part of the contribution, so I would encourage the authors to even consider adding an ablation condition demonstrating this is helpful for the performance of the system.

---

> > > ### Author Response · Authors · 2023-11-17
> > > **Look forward for your constructive response (Part 3)**
> > >
> > > **Q6. Ablation on Debug mechanism.**
> > >
> > > Table 4: Ablation on Debug mechanism.
> > > | Method GNSVR | GQA |
> > > | -------- | -------- |
> > > |  Baseline | 45.9 |
> > > |  w/o debug | 44.9 |
> > >
> > > The error-correction prompt contains the error message from Python interpreter and wrong code snippet. We prompt the LLM to correct the wrong code based on the error message from Python interpreter.
> > > We heuristically set the maximal number of debug iterations as 5. If the wrong code can be corrected within 5 iterations, we will keep it. Otherwise, it will be abandoned. (Details can be found in Module Generation section of Fig. 2)
> > > The errors mainly stem from two sources: one is basic syntax errors in Python code, such as indentation and variable name errors. The other source is some fundamental logical errors, such as mistakes made when setting variable types, like treating a variable that should be of the bool type as the string type.
> > > By observing the table above, we can conclude that the debug process can assist GNSVR with generating more useful modules to elevate performance and prevent elementary programming mistakes.
> > >
> > > > Q7. It would be informative to provide additional details on the generated modules for each task: how many are created for each task? How often does each one get used in solving test-cases? Do they each capture distinct concepts, or are some modules duplicates?
> > >
> > > **Q7. how many modules are generated for each tasks**
> > >
> > > **GQA** (13 modules)
> > > VERIFY_COLOR, VERIFY_ACTION, VERIFY_ATTRIBUTE, VERIFY_MATERIAL, VERIFY_OBJECT, COMPARE_ATTRIBUTE, COMPARE_COLOR, COMPARE_MATERIAL, CHOOSE_ATTRIBUTE, CHOOSE_COLOR, CHOOSE_MATERIAL, CHOOSE_DEPTH, CHOOSE_ACTION.
> > >
> > > **RefCOCO** (5 modules)
> > > FILTER_COLOR, FILTER_SHAPE, VERIFY_RELATION, FILTER_RELATION, SORT_SPATIAL
> > >
> > > **RAVEN** (4 modules)
> > > DETECT_COLOR, DETECT_SIZE, DETECT_SHAPE, SOLVER.
> > >
> > > **MEWL** (4 modules)
> > > DETECT_COLOR, DETECT_MATERIAL, DETECT_SHAPE, SOLVER.
> > >
> > > We further take GQA as an example to exhibit the percentage of top-5 most-used new modules in GQA in table 5.
> > >
> > > Table 5: Percentage of top-5 most-used new modules in GQA.
> > > |Task|VERIFY_ATTRIBUTE |CHOOSE_ATTRIBUTE|VERIFY_COLOR|COMPARE_ATTRIBUTE|VERIFY_MATERIAL|
> > > | -------  |  -------  | -------- | -------- | --- | --- |
> > > | GQA      | 14.1 | 10.8  | 6.9 | 5.9 | 3.6 |
> > >
> > > The data in the above table shows the proportion of the five most common new modules appearing in the generated high-level programs. Overall, 38.7% of all generated high-level programs use the newly generated modules (this 38.7% calculation includes other less common modules and excludes duplicate samples, such as a single high-level program containing multiple new modules). From these results, it can be seen that these newly learned modules can be widely applied to GQA, thereby helping GNSVR achieve good results on GQA.
> > >
> > > > Q8. (a) From the example prompts, it looks like a single function is given as an in-context example for step 1 and step 2, are these always COMPARE_SIZE and LOC, as shown in figures 15 and 16?
> > >
> > > **Q8. Details on in-context examples.** Indeed, there are more examples provided in the prompt of step 1 and step 2. We just show the first and representative ones in the Figure 15 and 16.
> > >
> > > > Q9. (b) For the inference in-context examples are these chosen randomly from the training tasks that found a successful program?
> > >
> > > **Q9. inference context examples.** Yes. They are randomly chosen from training samples which generated successful programs. These examples utilize various newly generated modules. Each new module is assigned at least three successful training examples.

---

> > > > ### Author Response · Authors · 2023-11-17
> > > > **Look forward for your constructive response (Part 4)**
> > > >
> > > > >Q10. (a) For the Raven task, my understanding is that the input is just a series of images. If this understanding is correct, how do you turn these images into a question? I am also quite curious to see the internals of the SOLVER generated module, is this module shared between Raven and MEWL, or does it employ distinct logic?
> > > >
> > > > **Q10. Details on Raven and MEWL.** Note that a visual reasoning task does **not** necessarily to use language as input. All we need is to prompt the LLMs to generate modules that recognize the patterns and solve the problem. In RAVEN, by prompting LLM, we can obtain DETECT_COLOR, DETECT_SHAPE, and DETECT_SIZE. The image is fed into these modules and the output is the color, shape, and size of the image. In this way, the input image is converted into a (color, shape, size) triplet. We provide LLM with ten examples from RAVEN train split to demonstrate how to deduce the pattern of these triplets. By observing few-shot demonstrations, we let LLM generate the SOLVER() module, which detects the pattern of input triplets from Problem Matrix and choose the most appropriate answer from the Answer Set. Therefore, the internal of the SOLVER() module is primarily based on judgment, used to identify the patterns of the input triples in the Problem Matrix, thereby finding the answer in the Answer Set. The workflow of RAVEN is shown in Fig. 5. As for MEWL, we employ a similar approach to handle it. One example is provided Fig. 6. Since MEWL and RAVEN have different patterns, the SOLVER() module is not shared between RAVEN and MEWL. Thus, it utilizes distinct logics.

---

> ### Comment · Reviewer_vHZ5 · 2023-11-20
>
> Thank you for the detailed responses! I've read through the author's comments, and after the rebuttal have a more favorable view of the paper. As such, I've updated my review accordingly. The consistent (though not terribly large) improvement over the baselines I suggested is an important mark in favor of the method, and tempers my fears about a fair comparison against ViperGPT/VisProg. The quality of the discovered modules having a reliance on a fairly large number of training examples is still a concern, but overall I would still support the paper's inclusion into the proceedings.

---

### Official Review · Reviewer_QVVg · 2023-11-06

**Soundness:** 3 good
**Presentation:** 4 excellent
**Contribution:** 2 fair
**Rating:** 5
**Confidence:** 5

**Summary:**

The authors introduce a Generative Neuro-symbolic Visual Reasoning Model (GNSVR) for vision-language tasks that involves creating, testing, and re-using neural "modules" i.e. neural network components that are steerable and solve well-defined abstraction and reasoning tasks. To evaluate these modules, a pre-trained LLM model is used to first determine whethere a new module is needed given a reasoning training set. Then, the LLM is asked to generate the function signature of the new module as well as an execution (reasoning) script to solve the given reasoning problem. The signature is used to generate a new module, which is evaluated against few samples to ascertain whether it is the right component. Once a generated module satisfies this criteria, it is added to the library of modules and the process is repeated. The empirical results show that GNSVR is able to develop novel reasoning modules, and utilize them successfully for a variety of vision language reasoning tasks.

**Strengths:**

Firstly, the paper is very well written and the GNSVR method is very well explained, with a healthy split between using the main paper and supplementary for splitting the key information versus additional information.

The empirical performance of GNSVR for transfer tasks, as well as the examples of the modules generated is quite impressive - it would justify the overall approach to growing modules for reasoning and their effectiveness for few-shot  generalization.

**Weaknesses:**

There are several components to the GNSVR framework but the paper provides no detailed analysis in the main paper of the importance of each of the components of the GNSVR framework.  While I think the framework overall is useful, the components are not all equally important to solve the reasoning problem and hence it is important to understand for future research on modular reasoning to understand what works and what doesn't, and if so why not.

How big of a role does "good initlialization" of the neural module operators plays?How important is defining the correct input and output format for a new module? How important is the selection of the few shot samples to evaluate a new module? (the authors say "We extracted 300 examples from GQA, 100 from RefCOCO, 10 from Raven, and 10 from MEWL based on experimental experience." - is the experience here just cherry picking for results or something else?) How important is the LLM capability to learn new modules? What role does the prompt play for the LLM in evaluating existing modules and creating new ones? Without detailed analysis to support answers to all these questions, the paper is limited in terms of explaining the method beyong just presenting a new method and showing empirical results.

Lastly, I would not suggest the authors not report results on the RAVEN dataset. As shown independently in [1] and [2] the dataset contains flaws in choice design which enables models to learn shortcuts to solve the RPM reasoning task. I would recommend the authors use the i-RAVEN dataset inroduced in [1] instead.

**References**

1. Hu, S., Ma, Y., Liu, X., Wei, Y. and Bai, S., 2021, May. Stratified rule-aware network for abstract visual reasoning. In Proceedings of the AAAI Conference on Artificial Intelligence (Vol. 35, No. 2, pp. 1567-1574).
2. Spratley, S., Ehinger, K. and Miller, T., 2020. A closer look at generalisation in raven. In Computer Vision–ECCV 2020: 16th European Conference, Glasgow, UK, August 23–28, 2020, Proceedings, Part XXVII 16 (pp. 601-616). Springer International Publishing.

**Questions:**

* Could the authors comment on why they chose to go with a neuro-symbolic approach versus a purely neural approach for defining the neural module scripts? Is it only for controllability, and if so can they comment on how a neural script (e.g. an LLM that abstracts the problem of generating the script and executing it) would compare? There have been approaches towards learning neural scripts and executing them dynamically at inference time for reasoning tasks in prior work e.g. [1]

* I think the module initialization and evaluation during generation in GNSVR is closely related to the research on automatic group discovery and using it for model design paradigm introduced in computer vision previously e.g. [2, 3]. It would make the related works section more comprehensive to include discussion on how GNSVR relates to this growing subfield of automatic evaluation and model design.


**References**

1. Rahaman, N., Gondal, M.W., Joshi, S., Gehler, P., Bengio, Y., Locatello, F. and Schölkopf, B., 2021. Dynamic inference with neural interpreters. Advances in Neural Information Processing Systems, 34, pp.10985-10998.
2. Vendrow, J., Jain, S., Engstrom, L. and Madry, A., 2023. Dataset interfaces: Diagnosing model failures using controllable counterfactual generation. arXiv preprint arXiv:2302.07865.
3. Gao, I., Ilharco, G., Lundberg, S. and Ribeiro, M.T., 2023. Adaptive testing of computer vision models. In Proceedings of the IEEE/CVF International Conference on Computer Vision (pp. 4003-4014).

---

> ### Author Response · Authors · 2023-11-17
> **Look forward for your further response (Part 1)**
>
> Thank you for the constructive comments.
>
> >Q1.  While I think the framework overall is useful, the components are not all equally important to solve the reasoning problem and hence it is important to understand for future research on modular reasoning to understand what works and what doesn't, and if so why not. How big of a role does "good initialization" of the neural module operators plays? How important is defining the correct input and output format for a new module? How important is the selection of the few shot samples to evaluate a new module? (the authors say "We extracted 300 examples from GQA, 100 from RefCOCO, 10 from Raven, and 10 from MEWL based on experimental experience." - is the experience here just cherry picking for results or something else?) How important is the LLM capability to learn new modules? What role does the prompt play for the LLM in evaluating existing modules and creating new ones? Without detailed analysis to support answers to all these questions, the paper is limited in terms of explaining the method beyond just presenting a new method and showing empirical results.
>
> **Q1. Ablation on each new module.**
>
> Thanks for raising these insightful questions, which help in understanding the importance of different components in the GNSVR. We randomly select 800 samples from GQA test-dev split to further investigate the effectiveness of different components of GNSVR. To better present all experimental results, all the mentioned ablation studies are organized into three sections below.
>
> **Q1-1: Ablation on Prompt Design.**
>
> Table 1: Ablation study of Prompt Design on GQA.
> | Method GNSVR | GQA |
> | -------- | -------- |
> |  Baseline | 45.9 |
> |  w/o input and output format | 43.2 |
> |  w/o good initialization | 41.8 |
> |  w/o existing modules in prompt for module making | 45.0 |
> |  w/o creating new modules   | 44.7 |
>
> We conducted a series of experiments to observe the impact of prompt design on the overall performance of GNSVR. Firstly, we removed the descriptions of input and output formats from the prompt. After removing these descriptions, the performance of GNSVR dropped by 2.7%. This is because, without clear guidance on input and output formats, the modules might output in the wrong format, leading to errors in subsequent parsing of the results. Furthermore, on top of removing the input and output format, we also removed some of the in-context examples and descriptions about module signatures from the prompt. The performance further declined. Since our method consists of three stages: module initialization, module generation, and module execution, where module initialization is the first step of our method. Without adequate module initialization as a foundation, the subsequent results are largely impacted. Therefore, we can see that without good initialization, our performance drops by 4.1%.
>
> Regarding the use of existing modules and creating new ones, from the table above, we can observe that not using the predefined modules from VisProg results in a 0.9% decrease in our performance. This demonstrates the robust module generation capability of GNSVR. Even without a series of predefined modules, our method can still build modules from scratch, solve problems, and the performance does not drop significantly. If we don't create new modules, then we are merely using the predefined modules. We can see that the result is 44.7%, which is 1.2% lower than our result of 45.9%. This performance gap highlights the effectiveness of the newly generated modules. By generating and using new modules, we can achieve better results.
>
> **Q1-2: Ablation on Sampling**
>
> In this section, we introduce our sampling strategy at first. Then, we conduct an experiment to showcase how the sampling methods will impact GNSVR performance. Subsequently, we investigate how the number of training samples affects our results of different tasks.
>
> **Sampling Strategy**
>
> Table 2: Ablation  on Sampling Strategy on GQA.
> | Method GNSVR | GQA |
> | -------- | -------- |
> |  Baseline | 45.9 |
> |  Random Sampling | 44.3 |
>
> Our sampling strategy: the GQA dataset contains five structural types: choose, logical, compare, verify, and query. These structural types inspired the idea of generating our new modules. Taking COMPARE_COLOR as an example, this newly generated module is generated to address questions related to color within the compare structural type. From the visualization of GQA, it is apparent that the query type can be addressed using existing VQA module from VisProg, and problems in the logical type can be decomposed into sub-problems of choose, compare, and verify types. Therefore, when selecting training samples, we randomly chose 100 samples each from the choose, compare, and verify types. Altogether, these three types comprise 300 samples, all sourced from the GQA train split. Hence, we are not cherry-picking our training samples; rather, we are selecting training samples based on the structural types of GQA.

---

> > ### Author Response · Authors · 2023-11-17
> > **Look forward for your further response (Part 2)**
> >
> > To explore the impact of sampling strategies on our experiment, we conducted an additional experiment with a random sampling of 300 samples, beyond our initial sampling strategy. In this setting, we randomly sampled 300 examples from the GQA train split. The performance was observed to be 44.3%, a decrease of 1.6% compared to 45.9%. This result suggests that a strategic sampling method can more effectively guide the LLM in generating more efficient modules for a given task. Relatively speaking, our method is robust in the face of choices in sampling strategies.
> >
> > **Number of Sampling Examples**
> >
> > Table 3: Number of Sampling Examples on GQA.
> > | # of samples | GQA |
> > | ------- | -------- |
> > | 60     | 44.5   |
> > | 120    | 45.3   |
> > | 300    | 45.9   |
> >
> > Table 4: Number of Sampling Examples on RefCOCO.
> > | # of samples | RefCOCO |
> > | ------- | -------- |
> > | 10     | 49.4   |
> > | 50     | 67.0   |
> > | 100    | 67.1   |
> >
> > Table 5: Number of Sampling Examples on RAVEN.
> > | # of samples | Center | L-R  | U-D  |
> > | ------------ | ------ | ---- | ---- |
> > | 5     | 46.5   | 37.2 | 39.8 |
> > | 10    | 80.1   | 67.6 | 69.1 |
> > | 20    | 80.1   | 67.6 | 69.1 |
> >
> > Table 6: Number of Sampling Examples on MEWL.
> > | # of samples | shape | color |  material  |
> > | ------------ | -------- | -------- | --- |
> > | 5     | 38.9   | 39.6 | 37.9 |
> > | 10    | 43.7   | 45.3 | 41.0 |
> > | 20    | 43.7   | 45.3 | 41.0 |
> >
> > We conduct a series of experiments to illustrate how the number of training samples influence the performance.
> >
> > In the GQA and RefCOCO datasets, if a small number of training samples are used, it's possible for the generated modules to overfit to certain samples, thereby reducing the generalization capability of the newly generated modules. Such overfitting in new modules can negatively impact the final results. Therefore, we can observe that when the number of samples is small, the performance of GNSVR is poorer. As the number of samples increases, the effectiveness of GNSVR improves. However, with a further increase in the number of samples, the performance gains of GNSVR tend to saturate.
> >
> > Regarding RAVEN and MEWL, since their patterns of change are limited, the number of few-shot samples selected is sufficient if it already covers all the variation patterns in RAVEN and MEWL. In other words, if the number of samples exceeds this threshold, there won't be any further improvement in the results; if it's below this threshold, the performance will decline. We selected 10 few-shot samples each in RAVEN and MEWL. As can be seen from the results in the table above, if the number of samples is equal to 5, there is a noticeable decrease in performance. This is because 5 few-shot samples are not enough to cover all the variation patterns of RAVEN or MEWL. If the number of samples is equal to 10 or 20, at this point, the few-shot samples are sufficient to encompass all possible variations. In this case, the same results are obtained.
> >
> > **Q1-3: Ablation on LLM's capability.**
> >
> > Table 7: Ablation on LLM's capability.
> > | GPT | GQA |
> > | -------- | -------- |
> > |  gpt-3.5-turbo-instruct | 45.9 |
> > |  gpt-3.5-turbo | 44.3 |
> >
> > By using a better LLM, our prompts can be better understood, and the LLM will generate higher quality modules. In this experiment, we compared the results of using gpt-3.5-turbo-instruct and gpt-3.5-turbo. Our experimental results show that better outcomes are achieved when using the more effective gpt-3.5-turbo-instruct. It is evident that the capabilities of the LLM influence the performance of GNSVR. As the abilities of LLMs continue to improve, so will the performance of GNSVR. Thanks to the flexibility of GNSVR, once a better LLM is available, we can easily switch to the latest LLM to achieve better results.
> >
> > >Q2 As shown independently in [1] and [2] the dataset contains flaws in choice design which enables models to learn shortcuts to solve the RPM reasoning task. I would recommend the authors use the i-RAVEN dataset introduced in [1] instead.
> >
> > **Q2. Experiments on the new RAVEN dataset.**
> > Thanks for the reminder of the potential shortcut issue of the RAVEN dataset proposed by Zhang *et al*. We have conducted further experiments on I-RAVEN dataset proposed by [1]. Following our setting in RAVEN (Zhang *et al*), we use 10 random-selected samples for learning new modules to handle the task and test the model on the testing set. As shown in table , our GNSVR is still able to handle the abstract reasoning task with high accuracy and data efficiency.
> >
> >  Table 8: Performance on the new I-RAVEN dataset.
> > | Dataset | Center | L-R |  U-D  |
> > | ------- | -------- | -------- | --- |
> > |LEN      |   56.4 | 44.2   |   44.2  |
> > |CoPINet  |   54.4 |  51.9  |   52.5  |
> > |SRAN     |  78.2  |70.1    |   70.3  |
> > | I-RAVEN |  85.2  |  74.6  |   75.4  |

---

> > > ### Author Response · Authors · 2023-11-17
> > > **Look forward for your further response (Part 3)**
> > >
> > > >Q3. Could the authors comment on why they chose to go with a neuro-symbolic approach versus a purely neural approach for defining the neural module scripts? Is it only for controllability, and if so can they comment on how a neural script (e.g. an LLM that abstracts the problem of generating the script and executing it) would compare? There have been approaches towards learning neural scripts and executing them dynamically at inference time for reasoning tasks in prior work e.g. [1]
> > >
> > > **Q3. Neuro-Symbolic approach VS a Purely neural approach.**
> > >
> > > We appreciate the suggestion to discuss both neuro-symbolic and purely neural approaches for neural modules. We posit that both purely neural-based approaches, such as [1], and neuro-symbolic models like our GNSVR, represent valuable explorations in enabling AI systems to abstract and solve reasoning problems through script generation and execution. In contrast to neuro-symbolic methods, purely neural approaches like [1] are end-to-end learnable solely from data, without dependence on pre-defined models or APIs. Conversely, our neuro-symbolic method, GNSVR, offers enhanced model transparency through explicit, modularized Python code snippets; 2) It facilitates the use of pre-defined models (e.g., perception models for classification) through explicit function calls; and 3) It demonstrates superior data efficiency in adapting to new tasks, as evidenced by using only 10 Raven examples to learn modules for reasoning tasks. We have included such discussion in the related work of the revised paper.
> > >
> > > [1] Rahaman, N., Gondal, M.W., Joshi, S., Gehler, P., Bengio, Y., Locatello, F. and Schölkopf, B., 2021. Dynamic inference with neural interpreters. Advances in Neural Information Processing Systems, 34, pp.10985-10998.
> > >
> > > >Q4. I think the module initialization and evaluation during generation in GNSVR is closely related to the research on automatic group discovery and using it for model design paradigm introduced in computer vision previously e.g. [2, 3]. It would make the related works section more comprehensive to include discussion on how GNSVR relates to this growing subfield of automatic evaluation and model design.
> > >
> > > **Q4. More Discussion on Related Work** Thanks for mentioning the related works[2,3]. While [2] and [3] focusing on improving pure neural network based model's performance and a different research area that automatically discover groups and use them for model design, we share similar interest on using LLMs and few-shot examples to improve AI models' performance. We have added such discussion in the related work of the revised paper.
> > >
> > > [2] Vendrow, J., Jain, S., Engstrom, L. and Madry, A., 2023. Dataset interfaces: Diagnosing model failures using controllable counterfactual generation. arXiv preprint arXiv:2302.07865.
> > > [3] Gao, I., Ilharco, G., Lundberg, S. and Ribeiro, M.T., 2023. Adaptive testing of computer vision models. In Proceedings of the IEEE/CVF International Conference on Computer Vision (pp. 4003-4014).

---

> > > > ### Author Response · Authors · 2023-11-21
> > > > **Looking forward to your post-rebuttal feedback**
> > > >
> > > > Dear Reviewer QVVg,
> > > >
> > > > Thanks again for your constructive suggestions and comments. As the deadline for discussion is approaching, we are glad to provide any additional clarifications that you may need.
> > > >
> > > > In our previous response, we have carefully studied your comments and added a lot more experiments and analysis to complement your suggestions.
> > > >
> > > > We hope that the provided new experiments and additional explanations have convinced you of the merits of our work. Please do not hesitate to contact us if there are other clarifications or experiments we can offer.
> > > >
> > > > Regards,
> > > > Authors

---

> > > > > ### Author Response · Authors · 2023-11-22
> > > > >
> > > > > Dear Reviewer QVVg,
> > > > >
> > > > > We sincerely thank you for your valuable feedback on our manuscript. Following your suggestions, we have enriched our paper with additional experiments and discussion about related work, which are now included in the revised manuscript.
> > > > >
> > > > > As the rebuttal period concludes in one day, we hope our efforts align with your expectations. If you find our response satisfactory, we would be grateful if you could consider revising your score as other reviewers.
> > > > >
> > > > > Thank you once again for your insightful guidance.
> > > > >
> > > > > Warm regards,
> > > > >
> > > > > Authors

---

> > > > > > ### Author Response · Authors · 2023-11-22
> > > > > >
> > > > > > Dear Reviewer #**QVVg**,
> > > > > >
> > > > > > We would like to thank you for your helpful feedback which has helped us improve the paper.
> > > > > >
> > > > > > We addressed reviewers' concerns in the author's responses, posted on the 17th of Nov 2023. We would be delighted if you could please take a look at our detailed responses so that we can address any remaining concerns before the end of the discussion phase.
> > > > > >
> > > > > > Sincerely,
> > > > > >
> > > > > > Authors of Submission 6843

---

### Official Review · Reviewer_CDRY · 2023-11-09

**Soundness:** 4 excellent
**Presentation:** 2 fair
**Contribution:** 3 good
**Rating:** 6
**Confidence:** 3

**Summary:**

This paper relies on the abilities of Large Language Models, but in effect extends them by attempting to enable reuse of code, as opposed to generating code from scratch.  They focus on the visual domain, where tasks include asking questions which require both visual analysis and symbolic reasoning.  They demonstrate state-of-the-art performance on several tasks

**Strengths:**

This paper makes a clear case for it's own contribution, and that contribution does appear to be valuable - visual learning is an important task, and the prohibitive cost of using SOTA large language models makes reuse of code appealing (although they explicitly don't use ChatGPT4 "due to the prohibitive cost", so maybe this is less of an argument than it would be otherwise).  The fact that they can show it's use on several domains and types of tasks is also appealing.

**Weaknesses:**

My main concern is that, based on the presentation, it seems that the authors took a lot of highly intricate API's for LLM's that large teams may have worked on and cobbled them together to solve a new task.  I refer to this section: "The success of our GNSVR relies on a set of pre-defined modules and APIs as the starting point. We utilize handcrafted modules from VisProg (Gupta & Kembhavi, 2022) as our initial components. Additionally, we incorporate several new APIs from ViperGPT to enhance module creation. We also include some new APIs from ViperGPT (Sur´ıs et al., 2023) for making new modules."  I appreciate their novelty in how they use these API's, but the ratio of insights of these authors vs of the authors of the API's appears insignificant.

I'm also not entirely convinced of the novelty of this paper.  I refer to "Iterative Disambiguation: Towards LLM-Supported Programming and System Design" (Pereira and Hartmann) and "Self-planning Code Generation with Large Language Models" (Jiang et al).  I don't think the fact that this is in the visual domain is enough to call it "novel", because there is virtually no engagement with visual modalities by the authors - as stated above, according to my understanding, they are using predefined modules which handle the interface between vision and language.

If I was given evidence against either of the two above claims - that all of the work (particularly the "visual reasoning" work) is being done by existing tools, or that the paper is not fundamentally using LLM's in a novel way - I would be happy to increase my score.

**Questions:**

Please clarify exactly where you created novel algorithms or ideas.  If any of those ideas require more than iterative prompting, please state so explicitly.

---

> ### Author Response · Authors · 2023-11-17
> **Look forward for your further feedback (Part 1)**
>
> We appreciate the reviewer for the detailed comments and insightful suggestions.
>
> >Q1. The prohibitive cost of using SOTA large language models makes reuse of code appealing (although they explicitly don't use ChatGPT4 "due to the prohibitive cost", so maybe this is less of an argument than it would be otherwise).
>
> **Q1. Computational Efficiency.** Thanks for mentioning the compute efficiency of our modularized design and module reusage. We have calculated the average token number of our GNSVR model and the ViperGPT that has no module reusage mechanism when calling the LLMs. The averaged generated token number is shown in Table 1. It can be seen that our GNSVR's solutions are shorter and more efficient. This result is updated in the revised paper. This approach is particularly advantageous when calling expensive APIs from the OpenAI GPT family.
>
> Table 1: Average token number of generated solutions.
> | Methods | GQA | RefCOCO |
> | -------- | -------- | -------- |
> | ViperGPT-Instruct     | 153.7     |  109.1  |
> | Ours-Instruct     | 62.3      |   54.4   |
>
>
> >Q2. My main concern is that, based on the presentation, it seems that the authors took a lot of highly intricate API's for LLM's that large teams may have worked on and cobbled them together to solve a new task. I refer to this section: "The success of our GNSVR relies on a set of pre-defined modules and APIs as the starting point. We utilize handcrafted modules from VisProg (Gupta & Kembhavi, 2022) as our initial components. Additionally, we incorporate several new APIs from ViperGPT to enhance module creation. We also include some new APIs from ViperGPT (Sur´ıs et al., 2023) for making new modules." I appreciate their novelty in how they use these API's, but the ratio of insights of these authors vs of the authors of the API's appears insignificant.
>
> **Q2. About Reliance of Existing Modules.**
> We agree that our GNSVR relies on some pre-defined modules as the initial start point to learn to generate new modules. However, we want to highlight that the novelty of GNSVR is **not** *"taking a lot of highly intricate API’s for LLMs to work on and cobble them together to solve a new task"*. Instead, our novelty lies in **growing** and **reusing** the established modules that are learnt from the training set. The modules learnt by GNSVR can be applied to different domains, including 1). other instances of the visual reasoning task; 2). instances of a new reasoning task; and 3). adapting to new reasoning tasks by observing only a few training examples. Please refer to the **G2** of the **general responses** for further explanation.
>
> >Q3. I'm also not entirely convinced of the novelty of this paper. I refer to "Iterative Disambiguation: Towards LLM-Supported Programming and System Design" (Pereira and Hartmann) and "Self-planning Code Generation with Large Language Models" (Jiang et al). I don't think the fact that this is in the visual domain is enough to call it "novel", because there is virtually no engagement with visual modalities by the authors - as stated above, according to my understanding, they are using predefined modules which handle the interface between vision and language.
>
> **Q3. About the paper's novelty compared with existing works.**
> Thanks for reminding related works[1,2]. We have added such revision in the related work section of the revised paper. In [1], Pereira and Hartmann used LLMs to progressively enhance and specify system subcomponents, empowering users to develop versatile programs through a systematic iterative disambiguation method. In [2], Jiang *et al* learned- to generate code with LLMs, which involves a planning phase for outlining solution steps and an implementation phase for generating code.
> Besides the **dense engagement with the visual modalities** as input, our GNSVR is different from them in **modularization** of code snippets for better **module expansion** and **module reusage**. These unique differences make our GNSVR model to have new capabilities like **growing** new modules to handle other instances of the visual reasoning tasks VQA, image grounding, RAVEN and MEWL and **reusing** these new modules in new tasks like image editing and knowledge tagging. Such modularization also offers better computational efficiency (See **Q1**) and model transparency with high-level module abstraction (See the *Generated Program* in Fig. 3, 5 and 6 for an example). Please refer to the **G2** of the **general responses** for further explanation.
> [1]. Iterative Disambiguation: Towards LLM-Supported Programming and System Design (Pereira and Hartmann).
> [2]. Self-planning Code Generation with Large Language Models (Jiang et al).

---

> > ### Author Response · Authors · 2023-11-17
> > **Look forward for your further feedback (Part 2)**
> >
> > >Q4. If I was given evidence against either of the two above claims - that all of the work (particularly the "visual reasoning" work) is being done by existing tools, or that the paper is not fundamentally using LLM's in a novel way - I would be happy to increase my score.
> >
> > **Q4-1. Evidence for all the work is NOT being done by existing tools.** We argue that our GNSVR model does **not** simply adopt all the tools for a visual reasoning task. Instead, it learns to make new tools (**grow modules**) and reusing these new tools with existing tools to handle a new task (**reuse modules**). Please refer to **Q2** and **Q3** for a detailed explanation.
> >
> > **Q4-2. Evidence for the paper is fundamentally using LLM’s in a novel way.** Our way is fundamentally novel to use LLMs to **grow** and **reuse** new modules. While existing works use LLMs to **solve each task instance independently without reusage**, our novel modularization design offers several benefits, including 1). better performance by examining the new modules from the given training examples; 2). efficient module reusage and transfer on other tasks; 3). better computation efficiency. We also show how we introduce a general and novel framework to use LLMs in **G2** of the **General Response**.
> >
> > >Q5. Please clarify exactly where you created novel algorithms or ideas. If any of those ideas require more than iterative prompting, please state so explicitly.
> >
> > **Q5. Clarification of Novelty.** As detailed explained in **G2** of the **general response**, GNSVR's core innovation lies in develop a **novel** and **general** framework to efficiently **generate** and **reuse** modules for visual reasoning tasks, even with limited training data. As shown in **G1** of the **general response**, our novelty to **grow** and **reuse** new modules is also well recognized by other reviewers **(QVVg,vHZ5,JpcA)**.

---

> > > ### Comment · Reviewer_CDRY · 2023-11-21
> > > **Thank you for your response**
> > >
> > > I am still not entirely convinced that the "visual reasoning" component is the main contribution, but I understand better now the novelty of your specific modular approach to programming with LLM's, and agree that in general modular programming is going to become increasingly important in the era of LLM's.

---

> > > > ### Author Response · Authors · 2023-11-21
> > > >
> > > > Dear Reviewer #CDRY,
> > > >
> > > > Thanks again for acknowledging the novelty of our GNSVR model to **grow** and **reuse** modules for LLM based programming for reasoning. We appreciate your recognition of the importance of modular programming in the era of LLMs. If we have convinced you of the claim that **"the paper is fundamentally using LLMs in a novel way"**, could you kindly increase your rating score for the paper according to your original comment that *"If I was given evidence against either of the two above claims - that all of the work (particularly the "visual reasoning" work) is being done by existing tools, or that the paper is not fundamentally using LLM's in a novel way - I would be happy to increase my score."*?
> > > >
> > > > Regards,
> > > >
> > > > Authors

---

### Author Response · Authors · 2023-11-17
**General Response: contributions, novelty, new Experiments and paper revision (Part 1)**

**G1. Contribution Recognition.**

We extend our sincere gratitude to the reviewers for their time and effort in reviewing our paper. We are pleased to note that the reviewers have generally acknowledged the GNSVR's following contributions:
* **the idea of growing new modules is promising.** Using LLMs to augment these APIs in a task-specific manner is an interesting idea **(vHz5)**; the idea of continuously growing this library is innovative, and the method simple and elegant **(JpcA)**.
* **the idea of reusing modules is appealing.** The prohibitive cost of using SOTA LLMs makes reuse of code appealing **(CDRY)**. The empirical performance of GNSVR for transfer tasks, as well as the examples of the modules generated is quite impressive **(QVVg)**; the claim is compelling that the modules that GNSVR finds from the GQA and RefCOCO tasks can "transfer" for related tasks of image editing and knowledge tagging **(vHZ5)**; I am appreciative of the idea of generating modular modules to be used in a library of skills for future tasks **(JpcA)**.
* **the proposed GNSVR framework has diverse applications.** The fact that they can show it's use on several domains and types of tasks is also appealing **(CDRY)**. GNSVR utilize new modules successfully for a variety of vision language reasoning tasks **(QVVg)**. It has many potential avenues for future exploration **(vHZ5)**.

**G2. Our core novelty.**

As recognized in **G1** of the general response, GNSVR's core innovation lies in develop a **novel** and **general** framework to efficiently **generate** and **reuse** modules for visual reasoning tasks, even with limited training data. This capability is grounded in two key processes:
1. **Module Generation**:
   - ***Assessment of Reusability***: Initially, for a given visual reasoning task, GNSVR evaluates the applicability of existing modules to the task at hand.
   - ***Initiation and Testing of New Modules***: If existing modules are deemed inadequate, GNSVR initiates the creation of a new module with LLMs. This process involves transforming training instances into "test cases" to evaluate module performance.
   - ***LLM-Driven Code Generation***: Utilizing LLMs, GNSVR then **generates** code snippets for the new module. These snippets are specifically tailored to pass the defined "test cases", ensuring functionality and task alignment.
2. **Module Reusage**:
   - ***Modularized Program Transformation***: When faced with a new query of a visual reasoning task, GNSVR translates this query into a structured, step-by-step modularized program.
   - ***Execution with Established Modules***: The program is executed by **reusing** previously established modules, showcasing the model's ability to apply existing knowledge to new scenarios.
   - ***Flexibility and Adaptability***: The approach facilitates 1) the handling of different instances within the same visual reasoning task (VQA, referring expression comprehension), 2) the application to new reasoning tasks (image editing and knowledge tagging), and 3) rapid adaptation to entirely new tasks with minimal training examples (RAVEN and MEWL).

 GNSVR is a novel and general framework that not only shows efficiency and effectiveness in tackling visual reasoning tasks but also embodies a significant leap towards more **adaptable** and **intelligent** AI systems. **The model's proficiency in generating and reusing modules offers a robust framework for continuous learning and adaptation, mirroring human cognitive processes in problem-solving and knowledge application.**

**G3. Experiments in the revision.**

To address the reviewers’ questions and support our responses, we conduct the following experiments to support our claims and show the effectiveness of our GNSVR model.
- Further ablation study on different components including (1). *good initialization*, (2). *input and output format*, (3). *prompt without existing modules*, (4). *prompt without creating new modules*, (5). *sampling strategy*, (6). *training number*, (7). *without module learning*, (8). *without debug mechanism* **(QVVg,vHZ5,JpcA)**
- (9). Experiments on the new I-RAVEN dataset **(QVVg)**
- (10). Variants of VisProg and ViperGPT on RAVEN and MEWL **(JpcA)**
- (11). Computational efficiency comparison of our modularized GNSVR model and ViperGPT **(CDRY)**.

---

> ### Author Response · Authors · 2023-11-17
> **General Response: contributions, novelty, new Experiments and paper revision (Part 2)**
>
> **G4. Paper Revision.**
>
> Besides the experimental results, we have also revised the paper correspondingly (highlighted in blue):
> - include experimental results into the revised version of the paper;
> - discuss the related work[A,B,C,D,E] and their difference with our GNSVR model;
> - revise text description for technique details.
>
> [A]. Iterative Disambiguation: Towards LLM-Supported Programming and System Design". Pereira and Hartmann
> [B]. Self-planning Code Generation with Large Language Models. Jiang *et al*.
> [C]. Dynamic inference with neural interpreters. NeurIPS. 2021. Rahaman *et al*.
> [D]. Dataset interfaces: Diagnosing model failures using controllable counterfactual generation. 2023. Arxiv. *Vendrow et al*.
> [E]. Adaptive testing of computer vision models. 2023. ICCV. Gao *et al*.

---

### Author Response · Authors · 2023-11-20
**General response (update): thanks for all your comments and look forward to post-rebuttal feedbacks!**

Thanks again for all of your constructive suggestions, which have helped us improve the quality and clarity of the paper!

Since it has been over three days after our initial response to the reviewers' concerns and questions, we have not heard any post-rebuttal response yet.

Please don’t hesitate to let us know if there are any additional clarifications or experiments that we can offer, as we would love to convince you of the merits of the paper. We appreciate your suggestions.

Thanks!

---

### Meta-Review · Area_Chair_ugw8 · 2023-12-05

**Metareview:**

This work is daring in its ambitions, surprising in its simplicity, and impressive in its scope (I can think of no other system that has ever tackled both VQA and Ravens with the same architecture!).

The main strength is showing that a basic form of learning a library of visual reasoning modules can enhance VQA.

The first weaknesses is that the method is mainly just prompting the model to introspect on when it should define a new library function.

The second weakness is that the module learning doesn't actually help that much: For instance on RefCOCO, performance goes from 62.3->67.1, and on GQA, 43.3->45.9. (The reader might infer better quantitative improvements over ViperGPT, but those appear to be due to improvements in the base prompt, not from the module learning. Thank you reviewers for getting to the bottom of this and requesting the experiments that would expose this fact.)

Several reviewers mentioned issues with clarity and lack of details. I would encourage the authors to include more details on exactly what the system is learning (for instance, I can find no explanation of what "SOLVER" is doing on the Ravens problems).

**Justification For Why Not Higher Score:**

There is a major weakness with this work: despite showing an exciting new method and piloting several interesting and diverse applications, the method itself doesn't seem to quantitatively boost performance on the main VQA application.

**Justification For Why Not Lower Score:**

The idea is clever and simple enough that it could plausibly catch on in the community. It also seems widely applicable to visual reasoning tasks, ranging from VQA to Ravens.

---

### Decision · Program_Chairs · 2024-01-16

Accept (poster)